# Challenges on Cyclic Nucleotide Phosphodiesterases Imaging with Positron Emission Tomography: Novel Radioligands and (Pre-)Clinical Insights since 2016

**DOI:** 10.3390/ijms22083832

**Published:** 2021-04-07

**Authors:** Susann Schröder, Matthias Scheunemann, Barbara Wenzel, Peter Brust

**Affiliations:** 1Department of Research and Development, ROTOP Pharmaka Ltd., 01328 Dresden, Germany; 2Department of Neuroradiopharmaceuticals, Institute of Radiopharmaceutical Cancer Research, Research Site Leipzig, Helmholtz-Zentrum Dresden-Rossendorf (HZDR), 04318 Leipzig, Germany; m.scheunemann@hzdr.de (M.S.); b.wenzel@hzdr.de (B.W.); p.brust@hzdr.de (P.B.)

**Keywords:** positron emission tomography, cyclic nucleotide phosphodiesterases, PDE inhibitors, PDE radioligands, radiochemistry, imaging, recent (pre-)clinical insights

## Abstract

Cyclic nucleotide phosphodiesterases (PDEs) represent one of the key targets in the research field of intracellular signaling related to the second messenger molecules cyclic adenosine monophosphate (cAMP) and/or cyclic guanosine monophosphate (cGMP). Hence, non-invasive imaging of this enzyme class by positron emission tomography (PET) using appropriate isoform-selective PDE radioligands is gaining importance. This methodology enables the in vivo diagnosis and staging of numerous diseases associated with altered PDE density or activity in the periphery and the central nervous system as well as the translational evaluation of novel PDE inhibitors as therapeutics. In this follow-up review, we summarize the efforts in the development of novel PDE radioligands and highlight (pre-)clinical insights from PET studies using already known PDE radioligands since 2016.

## 1. Introduction

This follow-up of our first review in 2016 [1] aims to report on (I) novel radioligands for imaging of cyclic nucleotide phosphodiesterases (PDEs) with positron emission tomography (PET) and (II) recent (pre-)clinical insights from PET studies using already known PDE radioligands. The biological and pathophysiological background of PDEs has previously been summarized [1] and thus, this review will be focused only on the current radiotracer development as well as the latest research findings. Briefly, PDEs are a class of intracellular enzymes that are expressed throughout the body. They are encoded by 21 genes and divided into 11 families that are subdivided into various subfamilies with different isoforms. Their central role is to hydrolyze the second messenger molecules, cyclic adenosine monophosphate (cAMP) and/or cyclic guanosine monophosphate (cGMP) and, therefore, to regulate the intracellular levels as well as the signaling cascades of these cyclic nucleotides. Several PDE inhibitors have been developed as therapeutics for treatment of various diseases like neurological, cardiovascular, immune or inflammatory disorders, cancer, and metabolism. However, the link between altered PDE expression and/or activity and pathophysiological effects often remains unclear. Hence, in vivo imaging and quantification of PDEs with appropriate radioligands for PET is commended as an important research and translational tool in related (pre-)clinical investigations.

To the best of our knowledge, there is still a lack of PET radioligands for the PDE families 3, 6, 8, 9, and 11. Herein, we will review novel radioligands as well as current results from (pre-)clinical PET studies related to the PDEs 1, 2A, 4, 5, 7 and 10A.

## 2. PDE1 Radioligands

PDE1 is one of dual-substrate specific PDEs that hydrolyses both cyclic nucleotides cAMP and cGMP. The special characteristic of the PDE1 family is the calcium/calmodulin-dependent regulation of the enzyme activity. Hence, PDE1 is suggested as an integration point for intracellular calcium and cyclic nucleotide signaling cascades [2]. The PDE1 enzyme is encoded by the three genes *PDE1A*, *PDE1B* and *PDE1C*, which are all expressed in the brain where PDE1B represents the most abundant PDE1 isoform. In the periphery, PDE1A is predominant in the kidneys, while PDE1C is highly expressed in the heart [3]. It is presumed, that PDE1-specific inhibitors might be suitable for the treatment of various neuropsychiatric and neurodegenerative diseases, like attention deficit hyperactivity disorder, depression, Parkinson’s disease and Huntington’s disease [4,5,6,7].

The first ^11^C- or ^18^F-labeled radioligands for PET imaging of PDE1 have been claimed in a patent in 2011 [8] as already reviewed by Andrés et al. [9]. However, it seems that further publications regarding subsequent biological investigations of these PDE1 radioligands are up to now not available. At the international symposium of functional neuroreceptor mapping of the living brain (NRM) in 2018, Kealey et al. [10] reported on the preliminary in vivo evaluation of the novel PDE1-specific radioligand (±)-[^11^C]**PF-04822163** ([^11^C]**1**, see Scheme 1). Out of a series of quinazoline-based PDE1-selective inhibitors developed by Humphrey et al. [11], compound **1** proved to be the most potent candidate (Scheme 1) and was thus selected for ^11^C-labeling. Starting from the corresponding phenol precursor, [^11^C]**1** has been synthesized by *O*-methylation using [^11^C]CH_3_I [10] (Scheme 1).

PET imaging studies in rats showed high initial brain uptake of [^11^C]**1** with a peak standardized uptake value (SUV) of 7 at 1 min post injection (p.i.), followed by a fast washout [10]. In the target region striatum, only slightly higher accumulation of [^11^C]**1** compared to the reference region cerebellum was observed, indicating high non-specific binding. Metabolite analysis revealed low in vivo stability with 45% and 30% of intact [^11^C]**1** in plasma at 5 min and 10 min p.i., respectively, while a single polar radiometabolite was detected [10]. Kealey et al. suggested that the fast degradation of [^11^C]**1** causes the rapid brain clearance, but further studies to investigate the radioligand kinetics and specificity of binding are required [10]. Overall, the short brain retention and low metabolic stability might limit the applicability of [^11^C]**1** for PET imaging of the PDE1 enzyme in the brain.

## 3. PDE2A Radioligands

The cAMP and cGMP dual-substrate specific PDE2A enzyme is abundantly expressed in the brain, particularly in structures of the limbic system [12,13], indicating an important role in the pathophysiology of neurodegenerative and neuropsychiatric diseases like Alzheimer’s disease, schizophrenia and dementia [14,15].

Since 2016, only one novel ^18^F-labeled radioligand for PET imaging of PDE2A, [^18^F]**BIT1** ([^18^F]**2**, see Figure 1), has been reported by our group [16]. This benzoimidazotriazine-based compound is a result of medicinal chemistry studies [17] to develop a PDE2A-specific radioligand with improved metabolic stability compared to our previous series of pyridoimidazotriazine derivatives with [^18^F]**TA5** ([^18^F]**3**, Figure 1) as the most potent candidate [1,18]. The structural modification of compound **3** led to our novel derivative **2** with a 16-fold decreased selectivity over PDE10A, which is also highly expressed in the striatum [3], while keeping the high PDE2A inhibitory potency [16,17] (Figure 1).

The in vitro metabolic stability of **2** has been proven by incubation with mouse liver microsomes and thus, this compound was selected for ^18^F-labeling starting from the corresponding nitro precursor (see Scheme 2). Metabolism studies in mice revealed sufficient stability with 78% of intact [^18^F]**2** in the brain at 30 min p.i. [16] demonstrating that the structural modifications resulted in a significantly improved in vivo stability compared to [^18^F]**3** (10% intact radioligand in the mouse brain at 30 min p.i. [18]). In PET imaging studies with mice, [^18^F]**2** showed good brain penetration with a SUV of 0.7 at 5 min p.i. However, a homogenous and non-specific activity distribution in the whole brain was observed which is consistent with the in vitro autoradiography on pig brain sections indicating high off-target binding of [^18^F]**2** [16]. Thus, we are currently working on further derivatives with enhanced PDE2A specificity and high metabolic stability.

Besides the ongoing efforts of our group to develop a suitable PDE2A radioligand, Chen et al. [21] reported on the pre-clinical evaluation of the already known and highly PDE2A-selective [^18^F]**PF-05270430** ([^18^F]**4**, see Figure 1), developed by Pfizer Inc. [14,19], in non-human primates. Briefly, [^18^F]**4** has previously been evaluated in healthy volunteers showing its suitability for PET imaging of PDE2A in the human brain [1,22,23]. The aim of the PET study in monkeys, performed in parallel, was to verify whether [^18^F]**4** is an appropriate radioligand for target occupancy studies in the assessment of novel PDE2A inhibitors as therapeutics [21]. Chen et al. performed the radiosynthesis of [^18^F]**4** by two different methods using (I) tetra-*n*-butylammonium [^18^F]fluoride ([^18^F]TBAF) in *tert*-amyl alcohol according to Zhang et al. [19] or (II) K^+^/[^18^F]F^−^/K_2.2.2_-carbonate complex as described by Morley et al. [24] but in *N*,*N*-dimethylformamide (DMF) instead of dimethyl sulfoxide (DMSO) [21] (see Scheme 3).

In vivo metabolism studies in rats revealed 30–46% and 93–95% of intact [^18^F]**4** at 30 min p.i. in plasma and striatum, respectively [21]. Chen et al. reported that most of the more polar radiometabolites are not supposed to enter the brain. However, only the results from analysis of the rat striatum (caudate nucleus and putamen) are provided. In rhesus monkeys, interanimal differences in metabolism have been described with 71%, 35%, and 55% of intact radioligand in plasma at 30 min p.i. [21]. Furthermore, it is stated that the results of the radiometabolite analysis of rat and monkey plasma are comparable indicating no significant species differences in the in vivo degradation of [^18^F]**4**. Baseline PET imaging studies in rhesus monkeys showed high activity uptake in the PDE2A-rich brain regions caudate nucleus and putamen as well as fast clearance from the whole brain [21]. Dose-dependent blocking of [^18^F]**4** uptake was achieved with a reduction of 73% in striatal binding potential at 2 mg/kg of **PF-05180999** [20] (Figure 1) as blocking agent indicating high specific binding of the radioligand. Target occupancy (TO) studies with **PF-05180999** and [^18^F]**4** in cynomolgus monkeys revealed a dose-dependent range of striatal TO values from 3% to 72% consistent with the mean plasma exposure of **PF-05180999** (40.2–240.0 ng/mL with an estimated target binding half maximal effective concentration of 69.4 ng/mL) [21]. Thus, [^18^F]**4** is stated as a suitable radioligand to measure the relationship between plasma concentration of the PDE2A inhibitor **PF-05180999** and its TO in the monkey brain by PET. Interestingly, a considerably higher brain uptake of [^18^F]**4** in cynomolgus monkeys compared to rhesus monkeys was observed while showing similar uptake patterns, kinetics and blocking effects [21]. Chen et al. discussed two possible reasons for that, namely higher PDE2A brain levels or slower metabolism in cynomolgus monkeys. However, metabolism studies of [^18^F]**4** have not yet been performed in cynomolgus monkeys and thus, the different brain uptake levels in these two monkey species remain unclear [21]. Based on the published results from the pre-clinical [19,21] and first in-human studies [22,23], [^18^F]**4** is regarded as particularly suitable radioligand for visualization and quantification of the PDE2A enzyme in the brain by PET.

## 4. PDE4 Radioligands

The intensively studied cAMP-specific PDE4 family is encoded by the four genes *PDE4A*, *B*, *C*, and *D* [25,26]. Out of these, PDE4B and PDE4D are the most abundantly expressed PDE4 mRNAs in the periphery while in the brain PDE4B is the main present isoform [3,27]. Alterations of PDE4 activity are related to disorders of the neurological, immune and inflammatory system, particularly depression, Alzheimer’s disease, chronic obstructive pulmonary disease and asthma [27,28,29,30,31].

### 4.1. Recent [^11^C]Rolipram Positron Emission Tomography (PET) Studies

In various (pre-)clinical studies, the well-known and non-subtype-specific PDE4 radioligand [^11^C]**rolipram** [32,33] (see Figure 2) has been demonstrated to be valuable for in vivo evaluation of the enzyme and the related cAMP signaling pathways by PET [9,34,35,36,37].

Accordingly, a lot of new insights from PET studies with [^11^C]**rolipram** have been published between 2017 and 2020. Fujita et al. [43] reported on a ~20% decrease of [^11^C]**rolipram** binding in the brain of unmedicated patients with major depressive disorder (MDD) compared to healthy controls indicating a significantly impaired cAMP signaling. Treatment with a selective serotonin reuptake inhibitor (SSRI) resulted in an increased uptake of [^11^C]**rolipram** of about 12% demonstrating a clear tendency to normalize cAMP signaling [43]. These results correspond to earlier findings and further support the importance of cAMP signaling for the etiology of depression. However, it has been found that there is no correlation between [^11^C]**rolipram** binding and either severity of baseline symptoms nor improvement of depressive or anxiety symptoms during SSRI treatment raising questions about the role of cAMP signaling in MDD. Fujita et al. [43] discussed several reasons for that, namely interactions between age and gender and the cAMP cascade as well as the heterogeneity of MDD and/or the target due to the fact that [^11^C]**rolipram** binds non-specifically to all four PDE4 subtypes. With regard to the latter, maybe a PDE4B-specific radioligand would be more useful to clarify the link between cAMP signaling and depression because of the high expression levels of the PDE4B isoform in the brain.

The first clinical PET study using [^11^C]**rolipram** to investigate the role of PDE4 in Parkinson’s disease (PD) was published in 2017 [44]. In L-DOPA-treated PD patients with no concurrent diagnosis of mild cognitive impairment or dementia, a 5–32% decrease of the [^11^C]**rolipram** volume of distribution (V_T_) has been observed compared to healthy controls. The reduction of V_T_ was significant in the subcortical nuclei, striatum, thalamus, hypothalamus and frontal cortex regions indicating impairment of PDE4 expression that correlated with working memory deficits in these PD patients [44]. Nevertheless, disease duration- and age-related changes in PDE4 levels need to be further investigated because of the small and wide cross-sectional cohort of PD patients and healthy controls [44]. Notably, in a previous study [45] Niccolini et al. reported on a decreased PDE10A expression in the striatum and pallidum of patients with moderate/advanced PD. Referring to that, a link between reduced PDE10A and PDE4 levels with the manifestation of motor and cognitive symptoms, respectively, in PD is proposed [44].

Interestingly, a recent [^11^C]**rolipram** PET study [46] revealed a significantly higher [^11^C]**rolipram** V_T_ in PD patients with excessive daytime sleepiness (EDS) compared with patients without EDS but not compared to healthy controls. This increased [^11^C]**rolipram** V_T_ was specifically found in brain regions regulating the sleep-awake cycle, like caudate nucleus, hypothalamus, hippocampus and limbic striatum [46]. The PDE4 upregulation is suggested to be not caused by treatment with dopamine agonists, because there was no difference in the medication between the groups of PD patients in this study [46]. These results indicate that EDS in PD is associated with increased PDE4 expression in related brain regions or that altered PDE4 expression might represent a possible mechanism causing the pathophysiology of EDS in PD [46]. Modulation of PDE4 in PD with EDS is suggested to be a suitable pharmacological target to enhance daytime sleepiness and behavioral changes related to sleep deprivation. However, further studies in larger cohorts of patients and healthy controls are required to understand the relationship between altered PDE4 expression and PD as well as sleep dysfunction in PD [46].

In 2018, a [^11^C]**rolipram** PET study explored the blockade of PDE4 in the healthy human brain in vivo by means of enzyme occupancy investigations [47]. For that purpose, the non-subtype-selective PDE4 inhibitor **GSK356278** [41] (see Figure 2) was used as blocking agent. The [^11^C]**rolipram** V_T_ was significantly decreased at 3 h, but not at 8 h after a single oral dose of **GSK356278** (14 mg) indicating good brain penetration and PDE4 binding of this inhibitor [47]. Based on the **GSK356278** plasma concentration and the [^11^C]**rolipram** free-plasma fractions (fp), a mean PDE4 occupancy of 49% and 19% at 3 h and 8 h post-dose, respectively, was assessed. There was no evidence for indirect pharmacokinetics of **GSK356278** in the human brain, suggesting that the estimated in vivo half maximal effective concentration (EC_50_(PDE4) = 46 ng/mL) is useful to calculate PDE4 occupancy and to determine optimal doses of **GSK356278** for future clinical development [47].

Another PDE4 occupancy study in non-human primates using [^11^C]**rolipram** PET has been reported by Takano et al. in 2018 [48]. The aim was to investigate whether **roflumilast** [42,49] (see Figure 2) enters the brain and demonstrates specific target occupancy. **Roflumilast** is the first PDE4 inhibitor that has been approved for treatment of severe chronic obstructive pulmonary disease (COPD) [50]. In **roflumilast**-treated rhesus monkeys, a reduced [^11^C]**rolipram** accumulation in all brain regions, a faster washout and a decreased V_T_ have been observed [48]. These results indicate that **roflumilast** is brain-penetrant and shows a dose-dependent PDE4 binding. The estimated PDE4 occupancy was <50% even at a dose of 200 µg/kg. Thus, Takano et al. stated that higher doses of **roflumilast** would be needed to confirm whether in vivo maximal PDE4 occupancy as measured with [^11^C]**rolipram** can reach 100% [48]. Apart from that, the obtained data may suggest that 30–40% of in vivo brain PDE4 occupancy, which corresponds to the clinical dose of **roflumilast**, might be an acceptable level of PDE4 inhibition not to induce severe nausea and emesis [48]. Further (pre-)clinical studies are required to investigate the relationship between **roflumilast** treatment, PDE4 occupancy in the brain, effects on cognitive domains, and occurrence of nausea and emesis [48]. For that purpose, Takano et al. suggested that PET studies using subtype-specific PDE4 radioligands instead of [^11^C]**rolipram** would be more effective to understand the function of the PDE4 isoforms [48].

In 2019, Ooms et al. [51] reported on a [^11^C]**rolipram** PET study to investigate the interaction between PDE4 and the disrupted in schizophrenia protein 1 (DISC1) in a DISC1 locus impairment mouse model (DISC1 LI). Disruption of the scaffold protein DISC1 leads to mental manifestations, like mood symptoms [52,53]. DISC1 is suggested as a potential PDE4 inhibitor and it is thus feasible that the absence of DISC1 would increase PDE4 activity [51,52,53]. PET imaging in DISC1 LI mice revealed a 41% higher [^11^C]**rolipram** V_T_ and a significant increase in [^11^C]**rolipram** V_T_/fp of 73% compared to wild-type mice [51]. These results are in good agreement with the ex vivo PDE4 enzyme activity assay, where a 25–50% increased PDE4 activity in brain homogenates of DISC1 LI mice has been shown [51]. Overall, this study demonstrates that [^11^C]**rolipram** PET is suitable to measure protein-protein interactions between DISC1 and PDE4 in vivo, that might also be interesting to be examined in further neuropsychiatric disorders in which DISC1 is dysregulated, including Huntington’s disease [51].

A recent clinical PET study in patients with the McCune-Albright syndrome (MAS) revealed a higher [^11^C]**rolipram** uptake in bones affected by fibrous dysplasia, as identified by [^18^F]**NaF** bone PET scans, compared to unaffected bones of MAS patients and healthy controls [54]. The genetic mutation in MAS leads to dysregulation of the cAMP pathway-associated G-protein and thus, to increased cAMP signaling and PDE4 activity. These results indicate that PDE4 upregulation is co-localized with areas of active bone remodeling in fibrous dysplasia. However, displacement studies are needed to verify whether the increased [^11^C]**rolipram** uptake is caused by specific PDE4 binding [54].

### 4.2. Novel PDE4 (B, D) Radioligands Apart from [^11^C]Rolipram

In 2017, Zhang et al. [55] reported on the novel PDE4B-preferring radioligand [^18^F]**PF-06445974** ([^18^F]**5**, see Figure 3). The aims of this work were to develop (I) a brain-penetrant PDE4B-specific inhibitor for treatment of brain diseases to enhance the therapeutic efficacy while suppressing/avoiding gastrointestinal side effects which are suggested to be partially associated with inhibition of the PDE4D isoform, and (II) a PDE4B-selective radioligand for accurate target occupancy as well as enzyme density and biodistribution determinations by PET.

Out of a tricyclic pyrrolopyridine series [55], compound **5** showed highest inhibitory potency towards PDE4B in the subnanomolar range, good to moderate selectivity over the isoforms PDE4A/C/D (Figure 3) and a favorable pharmacokinetic profile. Furthermore, ex vivo liquid chromatography–tandem mass spectrometry (LC-MS/MS) studies in rats and 129/B6 PDE4D KO mice revealed high brain permeability and excellent brain uptake of **5** which was significantly reduced after pre-administration of **rolipram** indicating high specific binding. Thus, **5** was selected for ^18^F-labeling starting from the corresponding nitro precursor [55] (see Scheme 4).

PET studies in cynomolgus monkeys revealed high accumulation of [^18^F]**5** in thalamus, followed by putamen and caudate nucleus, various cortical regions, and lower levels in cerebellum and hippocampus with peak SUVs of about three within less than 30 min p.i. in all brain regions [55]. This brain accumulation was significantly reduced after pre-administration of a structurally distinct PDE4B-preferring inhibitor (IC_50_(PDE4A/B/C/D) = 3/6/3.5/660 nM [55]) with an estimated PDE4B occupancy of ~92% [55]. In conclusion, these preliminary results indicate that [^18^F]**5** is a promising PDE4B-preferring radioligand for in vivo target occupancy measurements. Zhang et al. already announced further studies regarding metabolic stability, radiation dosimetry and additional PDE4B occupancy determination [55] which will certainly be followed with high interest.

For PET imaging of neuroinflammation, two novel ^11^C-labeled radioligands that are dual specific for the soluble epoxidase hydrolase (sEH) and PDE4 were reported in 2019 by Jia et al. [56]. Multi-target drugs for treatment of complex diseases are regarded to be more efficient than the traditional single-target approach because therapeutics often interact with multiple targets [60,61,62,63,64]. Both, sEH and PDE4, are critical enzymes in neuroinflammation and play an important role in the progression of various neurodegenerative diseases including Alzheimer’s disease [57]. Thus, Jia et al. selected the recently developed bioavailable sEH/PDE4 dual inhibitors **MPPA** (compound **6**) and a related derivative (compound **7**) for treatment of inflammatory pain as lead compounds [56,57] (see Figure 3). Starting from the respective phenol precursors, the sEH/PDE4 radioligands [^11^C]**6** and [^11^C]**7** have been synthesized in a home-built automated multi-purpose ^11^C-radiosynthesis module [56] (see Scheme 5).

Jia et al. reported that their radiosynthesis approach enabled the automated ^11^C-labeling reaction as well as purification and formulation for routine production suitable for preparation of clinical doses of the radioligands [56]. This facilitates future studies for the planned biological evaluation of [^11^C]**6** and [^11^C]**7**.

Very recently, Wakabayashi et al. [58] reported on the development of four highly PDE4D-specific radioligands, [^11^C]**T1650**, [^11^C]**T1660**, [^11^C]**T1953** and [^11^C]**T2525** ([^11^C]**8–11**, see Figure 3), for brain PET imaging of the enzyme especially related with neuropsychiatric disorders like major depression (MD). In contrast to the above discussed work of Zhang et al. [55], where specificity towards the PDE4B isoform was preferred, Wakabayashi et al. stated that selective PDE4D inhibition may preserve antidepressant effects while decreasing side-effect liability [58,65]. Thus, it seems that this interesting question requires further investigations to elucidate the correlation between selective pharmacological inhibition of PDE4B or PDE4D enzyme activity and related side effects. Out of a large series of compounds, the four pyridinyl core derivatives **8**, **9**, **10** and **11** have been selected for ^11^C-labeling based on their high inhibitory potencies for PDE4D and selectivity over PDE4B (Figure 3), moderate computed lipophilicities (cLogD_7.4_ = 2.7–3.9), favorable PET multi parameter optimization scores (MPO = 2.6–3.9) as well as sufficient peak uptake in mouse brain (SUV ≥ 1). Starting from the corresponding desmethyl precursors, the respective radioligands have been synthesized by *O*- or *N*-methylation using [^11^C]CH_3_I [58] (see Scheme 6).

PET studies in rhesus monkeys revealed high initial brain uptake of the four radioligands with peak SUVs of 3.3 to 5.1 at 4–5 min p.i. [58]. In plasma, [^11^C]**9** and [^11^C]**11** showed high stability while for [^11^C]**8** and [^11^C]**10** moderate stability was observed. After pre-administration of **rolipram** (0.1–0.5 mg/kg), no blocking effect was detected regarding [^11^C]**11** uptake and for [^11^C]**10** only little effect was achieved by faster washout while V_T_ and V_T_/fp increased. These results indicate no specific PDE4D binding of these two radioligands [58]. For [^11^C]**9**, an earlier and higher peak SUV (4.15 at 2.3 min vs. 3.52 at 5.5 min), faster washout as well as decreases in V_T_ and V_T_/fp by 37% and 45%, respectively, were observed under blocking with **rolipram** (1 mg/kg). The highest levels of [^11^C]**9** specific binding were detected in cortex and hippocampus and the estimated PDE4D occupancy by **rolipram** was 79% [58]. Little blocking effects for [^11^C]**8** were achieved with 48% and 41% decreased V_T_ and V_T_/fp, respectively, using **rolipram** (1 mg/kg). The [^11^C]**8** V_T_ was reduced by 37% after pre-administration of the PDE4D-specific allosteric inhibitor **BPN14770** [59] (3 mg/kg, Figure 3) with a slightly higher peak SUV (3.9 at 4 min vs. 3.28 at 5 min) and faster washout. Furthermore, a homogenous activity distribution under blocking with **BPN14770** was observed which might indicate the formation of brain-penetrating radiometabolites as reported for the rat brain with only about 60% of intact [^11^C]**8** at 30 min p.i. [58]. PDE4D occupancy using [^11^C]**8** was estimated with 63% and 93% at 0.2 mg/kg and 1 mg/kg of **rolipram**, respectively, and 63% at 3 mg/kg of **BPN14770**. Wakabayashi et al. concluded that due to very similar non-displaceable binding levels with both blocking agents, [^11^C]**8** shows PDE4D-specific binding [58]. For both radioligands, [^11^C]**8** and [^11^C]**9**, an atypical increase of the V_T_ over the scanning period were observed, hampering robust quantification of PDE4B in the brain by PET [58].

Although [^11^C]**9** showed higher metabolic stability in monkeys, Wakabayashi et al. selected [^11^C]**8** for a first PET study in two healthy volunteers [58]. In the human brain, a high uptake with peak SUV of 4.39 at 4 min p.i. was observed that slowly declined and plateaued from 30 min to 120 min (SUV ~2). In plasma, activity decreased by 71% from 60 min to 120 min indicating fast clearance while several radiometabolites have been detected with only 30% and 5% of intact [^11^C]**8** at 30 min and 120 min p.i., respectively. Thus, the higher uptake and slow washout in brain compared to plasma might reflect the formation of brain-penetrating radiometabolites, as already detected in rats and mentioned above. Moreover, the radiometabolite profiles of [^11^C]**8** in rat, monkey and human plasma were very similar [58]. Pre-administration of **BPN14770** (50 mg/kg) revealed a significant blocking effect with 30% and 33% decrease of SUV (at 60–120 min) and V_T_, respectively. The highest V_T_ reduction was observed in hippocampus and amygdala, and lowest decrease in basal ganglia, thalamus and cerebellum indicating specific PDE4D binding of [^11^C]**8** [58]. Nevertheless, the occurrence of brain radiometabolites limits the suitability of [^11^C]**8** for correct quantification of PDE4D in the brain by PET. These results also led to the exclusion of [^11^C]**9** for human PET studies due to the close molecular structure and the similar performance in monkeys compared with [^11^C]**8** [58]. In conclusion, none of the four novel PDE4D-selective radioligands turned out to be suitable for PET imaging of the PDE4D isoform in the brain due to high non-specific binding ([^11^C]**10** and [^11^C]**11**) and/or the formation of brain-penetrating radiometabolites ([^11^C]**8** and [^11^C]**9**) [58].

At the virtual meeting of the society of nuclear medicine and molecular imaging (SNMMI) in 2020, Telu et al. [66] reported on further seven PDE4D-specific radioligands (see Scheme 7 for [^11^C]**12**, [^11^C]**13** and [^18^F]**14**), for drug development and investigation of PDE4D regulation in neuropsychiatric disorders. All of the novel ligands showed high inhibitory potencies with IC_50_ values for PDE4D in the subnanomolar and low nanomolar rage as well as good selectivity towards PDE4B (exact data not published). The related radioligands have been synthesized starting from the corresponding desmethyl or iodonium salt precursors [66] (Scheme 7). 

PET studies in rhesus monkeys revealed that all of the radioligands readily entered the brain with fast washout. However, only [^11^C]**12** and [^11^C]**13** showed moderate brain uptake (peak SUV ~ 3) and considerable PDE4D-specific binding as determined under blocking with either **rolipram** or **BPN14770** [66]. Telu et al. stated that the observed V_T_ time stability for both radioligands indicate possible accumulation of radiometabolites in the brain but was thought sufficient for PDE4D quantification by PET since the normalized V_T_ improves towards the end of the scan period (120 min). Out of this compound series, [^11^C]**13** is suggested to be the most appropriate PDE4D radioligand and thus, possibly will be further investigated in target occupancy studies by PET [66].

## 5. PDE5 Radioligands

The cGMP-specific PDE5 enzyme is widely distributed in peripheral tissues like lung, bladder, penis, stomach, thyroid, pancreas, heart, and intestine where it regulates vascular smooth muscle contraction by controlling the intracellular cGMP concentration [2,3,67,68]. In the brain, the PDE5 expression levels are rather low [3,69]. Nevertheless, pharmacological inhibition of PDE5 activity in the brain is suggested to play an important role in cognition-related neural function and can promote beneficial effects on cognition and memory in both physiological and pathophysiological conditions [70]. PDE5 inhibitors are also gaining importance for treatment of neurodegenerative and neuropsychiatric disorders such as Alzheimer’s disease, epilepsy, stroke, depression and Huntington’s disease [69,70,71].

In 2017, Chekol et al. [72] reported on two novel PDE5-specific radioligands, [^11^C]**15** and [^18^F]**16** (see Figure 4), for evaluation of the enzyme expression and occupancy in lung, heart and brain by PET. 

Out of a series of pyridopyrazinone derivatives, compounds **15** and **16** showed highest inhibitory potency towards PDE5 (Figure 4) and were subsequently selected for radiolabeling. Starting from the corresponding desmethyl or tosylate precursors, [^11^C]**15** and [^18^F]**16** have been synthesized by *N*-methylation using [^11^C]CH_3_OTf or nucleophilic aliphatic substitution via the K^+^/[^18^F]F^−^/K_2.2.2_-carbonate complex [72] (see Scheme 8).

Biodistribution studies in mice revealed initial uptake of both radioligands at 2 min p.i. in kidneys, liver, lungs and intestine with hepatobiliary excretion [72]. Highest uptake was observed in lungs at all time points with fast washout. [^18^F]**16** showed a three- to fourfold higher % injected dose (%ID) in blood at 30 min p.i. compared to [^11^C]**15** which might be a result of circulating radiometabolites of [^18^F]**16**. For both radioligands, myocardial uptake and retention was negligible, consistent with the low PDE5 expression in the healthy heart. In contrast, myocardial uptake in transgenic mice with cardiomyocyte-specific bovine PDE5 overexpression (PDE5-TG) was significantly higher with a 10-fold and 16-fold increase for [^11^C]**15** and [^18^F]**16**, respectively. In the brain, [^11^C]**15** and [^18^F]**16** displayed relatively high uptake with 4%ID and 2.4%ID at 2 min p.i., respectively, while longer retention was observed for [^11^C]**15** [72]. Pre-treatment with **tadalafil** [78,79] (10 mg/kg; see Figure 4) revealed a significantly reduced lung retention by 32% and 70% for [^11^C]**15** and [^18^F]**16**, respectively. This result indicates that the accumulation of both radioligands in the lungs is only partially due to PDE5-specific binding, especially for [^11^C]**15**. A substantial **tadalafil** blocking effect was observed in PDE5-TG mice with 80% and 87% decreased myocardial retention for [^11^C]**15** and [^18^F]**16**, respectively, indicating PDE5-specific binding of both radioligands in the heart. Brain uptake of neither [^11^C]**15** nor [^18^F]**16** was reduced under blocking conditions because of the very limited brain penetration of **tadalafil** [72] demonstrating that this PDE5 inhibitor is not suitable as blocking agent for brain target investigations. Metabolism studies in mice revealed moderate to low in vivo stability of the radioligands with 30% and 5–7% of intact [^11^C]**15** and [^18^F]**16** in plasma at 30 min p.i., respectively. Negligible radiometabolite levels have been detected in the brain with about 90% of intact [^11^C]**15** and [^18^F]**16** at 30 min p.i. [72]. For PET studies in PDE5-TG mice, Chekol et al. selected [^18^F]**16** due to the higher myocardial uptake compared to [^11^C]**15**. In the PDE5-overexpressing heart, [^18^F]**16** showed a high peak SUV of 6.7 followed by a brief transient equilibrium between 2 and 10 min p.i.. This myocardial uptake was decreased rapidly under administration of **tadalafil** at 10 min after injection of the radioligand indicating reversible binding. Brain PET imaging has been performed in rats with [^11^C]**15** and [^18^F]**16** [72]. Both radioligands crossed the blood–brain barrier (BBB) with peak SUVs of 3–4 for [^11^C]**15** and 1.2–1.6 for [^18^F]**16** and were retained in the brain with a relatively homogenous distribution. Self-blocking with **16** revealed no significant difference in [^18^F]**16** uptake possibly due to the low PDE5 levels in the brain or non-saturable binding [72]. Thus, Chekol et al. suggested a lysosomal trapping of both radioligands that is supported by the high non-displaceable fractions in lung [72]. In vitro autoradiography with [^11^C]**15** and [^18^F]**16** on rat brain slices has been performed to further verify this assumption. A homogenous distribution of both radioligands in the grey matter was observed with moderate decrease of 45% and 35% for [^11^C]**15** and [^18^F]**16**, respectively, under self-blocking conditions (20 µM of each). This blocking effect was achieved to a much lesser extent using the non-selective PDE inhibitor **IBMX** (1 mM) or the PDE5-specific inhibitors **vardenafil** [78,80] (see Figure 4) and **tadalafil** (20 µM of each). Chekol et al. stated that these results correspond to the hypothesis of lysosomal trapping of both radioligands [72]. In conclusion, [^18^F]**16** is suggested as a suitable PDE5 radioligand for imaging and quantification of the enzyme expression in the myocardium of heart failure patients and for myocardial target occupancy studies to investigate by PET novel PDE5 inhibitors as potential therapeutics [72]. The observed high non-specific retention of both radioligands in the brain might limit their suitability for brain PET imaging of PDE5 [72].

For PET studies of PDE5 in the brain in particular, the INSERM branch at Clermont-Ferrand, France, developed in cooperation with our group the novel radioligand [^18^F]**17** [73] (see Figure 4). Based on our first PDE5 radioligand [^18^F]**ICF24027** [1,74] ([^18^F]**18**, Figure 4) reported in 2016, structural modifications have been done to improve the metabolic stability by introducing the fluorine-containing moiety in another position of the molecule [73,81]. Out of a series of PDE5-specific quinoline-based ligands, compound **17** is the most promising derivative (Figure 4). Since nucleophilic ^18^F-labeling of a secondary carbon is challenging, two strategies regarding an appropriate leaving group have been pursued, via the tosylate and the nosylate precursor [73] (see Scheme 9). Interestingly, highest radiochemical yields for automated radiosynthesis of [^18^F]**17** were achieved by using the nosylate precursor and [^18^F]TBAF with the addition of small amounts of water to the reaction mixture [73].

In vitro autoradiography on porcine brain slices revealed moderate binding of [^18^F]**17** in cerebellum, cortex and hippocampus [73] that are regions with identified PDE5 expression in the human brain [69]. Only a 10–25% decrease of [^18^F]**17** accumulation in all investigated brain areas has been achieved using **sildenafil** [78,79,80] (see Figure 4) or compound **17** (1 µM of each) as blocking agents indicating high non-specific binding of the radioligand [73]. We assume that the inhibitory potency of [^18^F]**17** might be too low due to the poor expression of PDE5 in the brain and thus, radioligands with subnanomolar potency might be needed for accurate PDE5 imaging in that regard. In biodistribution studies in mice, sufficient brain uptake of [^18^F]**17** with a peak SUV of 1.1 at 5 min p.i. and a fast washout have been observed. High initial [^18^F]**17** uptake was shown in lungs and heart with SUVs of 8.7 and 5.0 at 5 min p.i., respectively, consistent with the high PDE5 levels in these tissues [73]. Metabolism studies in mice revealed fast degradation of the radioligand with only 13% and 24% of intact [^18^F]**17** in plasma and brain at 30 min p.i., respectively, while high fractions of more polar radiometabolites have been detected in the brain [73]. In conclusion, [^18^F]**17** proved to be inappropriate for in vitro or in vivo imaging and quantification of the PDE5 enzyme due to high non-specific binding, low metabolic stability and the formation of brain-penetrating radiometabolites.

In 2019 and 2020, three novel ^11^C-labeled PDE5 radioligands, [^11^C]**19**, [^11^C]**20** and [^11^C]**TPN171** ([^11^C]**21**, see Figure 4) have been published by collaborating research groups from Hebei, China, and Indiana, USA [75,77]. Dong et al. [75] focused on the PDE5 enzyme as a promising non-beta amyloid-related target for treatment and imaging of Alzheimer’s disease and thus, [^11^C]**19** and [^11^C]**20** were generated to investigate PDE5 in the brain by PET. For that purpose, radioligands with subnanomolar potency and high target selectivity are required as mentioned above. Out of a series of novel naphthyridine and 1*H*-pyrroloquinolinone analogs [75,82], the most potent candidates **19** and **20** displayed picomolar inhibitory potency towards PDE5 and excellent selectivity over PDE6 (Figure 4), which shows high similarity in the catalytic site with PDE5 [82]. Besides, Xu et al. [77] reported on the development of [^11^C]**21** for PET imaging of PDE5 in the heart. Briefly, the 4(3*H*)-pyrimidine lead compound **21** has previously been developed by Wang et al. [76] as highly potent PDE5 inhibitor (Figure 4) for the treatment of pulmonary arterial hypertension and is currently being investigated in a phase II clinical trial.

The radioligands [^11^C]**19**, [^11^C]**20** and [^11^C]**21** were prepared by *O*- or *N*-methylation of the corresponding desmethyl precursors using [^11^C]CH_3_OTf in a home-built automated multipurpose ^11^C-radiosynthesis module [75,77] (see Scheme 10). This radiosynthesis approach enabled the robust production of [^11^C]**19**, [^11^C]**20** and [^11^C]**21** in good radiochemical yields and sufficient molar activities (Scheme 10) for future pre-clinical evaluation of these highly potential PDE5 radioligands [75,77].

## 6. PDE7 Radioligands

PDE7 is a cAMP-specific enzyme that is encoded by the two genes *PDE7A* and *PDE7B* [2,3,83]. PDE7A is expressed in high levels in spleen, heart, skeletal muscle and various immune cells and at low levels in the brain while PDE7B is predominantly distributed in striatal brain regions [3,84,85,86,87,88]. Although only little is known about the (patho-)physiological effects of the enzyme activity, selective PDE7 inhibitors are regarded as potent therapeutics for inflammatory and neurological disorders related with T-cell activation, multiple sclerosis, Parkinson’s disease, Alzheimer’s disease and addiction [89,90,91,92,93,94,95].

The first PDE7 radioligands [^18^F]**MICA-003** and [^11^C]**MICA-005** [1,96,97] ([^18^F]**22** and [^11^C]**23**, see Figure 5) have been reported by Thomae et al. in 2015, which failed for PET application due to the formation of brain-penetrating radiometabolites. In the last five years, only two further radioligands for imaging of the PDE7 enzyme in the brain have been published. Very recently, the tritiated PDE7B-specific radioligand [^3^H]**24** [98] (Figure 5) has been described by Chen et al., which is structurally related to [^18^F]**22** and [^11^C]**23**. The aim of this work was the fast evaluation of the distribution of novel PDE7B ligands in the rat brain by ex vivo LC-MS/MS screening to identify promising candidates as suitable radioligands for target occupancy studies. Out of six investigated highly potent PDE7B inhibitors, compound **24** (5 µg/kg) showed a striatal accumulation of 0.25%ID/g brain tissue and the highest striatum-to-cerebellum ratios of 2.4, 2.9 and 2.2 at 5 min, 30 min and 60 min p.i., respectively, indicating reasonable target specificity [98].

Thus, **24** has been selected for ^3^H-labeling to further validate its PDE7B-specific binding by ex vivo autoradiography on rat brain slices. At 10 min p.i., [^3^H]**24** (22 kBq/g, A_m_ = 3.5 GBq/µM; 2.7 µg/kg) displayed a striatum-to-cerebellum ratio of 2.6 consistent with the ex vivo LC-MS/MS results of unlabeled **24** [98]. In rat brain homogenates, a saturable PDE7B-specific binding of [^3^H]**24** has been observed, with an estimated dissociation constant (*K*_D_) of 0.8 nM and an enzyme density (*B*_max_) of 58 fmol/mg, which could be displaced under self-blocking conditions at 10 µM of **24** and in a dose-dependent manner by a structurally distinct blocking agent (*K*_i_(PDE7) = 120 nM) [98]. In conclusion, these preliminary results indicate that compound **24** is a very promising candidate regarding the development of a corresponding PET radioligand for in vivo visualization and quantification of the PDE7B enzyme in the brain. With that regard and although PDE7A levels in the brain are lower than for PDE7B [3,84,85], selectivity of **24** over the PDE7A isoform would also be of great interest.

The second novel PDE7 radioligand [^11^C]**MTP38** [99] ([^11^C]**25**, see Figure 5), which is structurally unrelated to [^18^F]**22** and [^11^C]**23**, has been developed by Obokata et al. in 2020. Based on a series of bicyclic triazolopyrazine derivatives (patent pending: WO 2018/038265), compound **25** showed favorable inhibitory potency towards PDE7A and PDE7B as well as high selectivity over all other PDEs (Figure 5) and various further off-targets. For the radiosynthesis of [^11^C]**25**, a two-step one-pot strategy has been applied starting from the corresponding bromo precursor [99] (see Scheme 11).

In vitro autoradiography on brain slices of rat and rhesus monkey revealed high accumulation of [^11^C]**25** in the striatum, which was significantly reduced by pre-incubation with either **25** (10 µM) or the PDE7 inhibitor **MTP-X** (4 µM; *K*_i_(PDE7) = 10 nM) indicating saturable and PDE7-specific binding of the radioligand [99]. In PET studies, [^11^C]**25** rapidly entered the brain with peak striatal SUVs of >4 at 1 min p.i. in rats and around 5 at 3–4 min p.i. in rhesus monkeys, respectively. In rats, a striatum-to-cerebellum ratio of 1.5 from 5 min to 60 min was estimated, which declined to 1.1 under blocking conditions, further supporting the specificity and time-stability of the striatal [^11^C]**25** binding. Additionally, a high [^11^C]**25** uptake in the olfactory epithelium was detected in both species. Brain clearance was slightly slower in rhesus monkeys than in rats but, generally, a fast washout was observed. After pre-treatment with **25** or **MTP-X**, the striatal activity levels in rats and rhesus monkeys were significantly decreased demonstrating high PDE7-specific binding of [^11^C]**25** in vivo consistent with the in vitro results [99]. Metabolite analysis revealed moderate in vivo stability of [^11^C]**25** in monkeys with 27% of intact radioligand in plasma at 90 min p.i. while one major and more polar radiometabolite was detected [99]. Striatal PDE7 occupancies of 53% and 87% at 30 mg/kg of **MTP-X** have been estimated in two monkeys, respectively [99]. Based on these promising pre-clinical results, Obokata et al. suggest [^11^C]**25** to be an appropriate radioligand for in vivo investigation of the PDE7 enzyme in the brain [99].

Very recently, the first-in-human PET study using [^11^C]**25** has been published by Kubota et al. [100]. PET scans in seven healthy volunteers displayed a high initial brain uptake of [^11^C]**25** with peak SUV values between 4 and 8 [100]. The activity in the non-target region cerebellar cortex decreased rapidly while a slow washout from PDE7-rich brain areas like pallidum and putamen was observed indicating specific binding of the radioligand. The highest V_T_ and non-displaceable binding potential (BP_ND_) values of [^11^C]**25** were assessed in the pallidum (4.2 and 0.55) and the putamen (3.9 and 0.46) followed by pons, thalamus and caudate nucleus, which is consistent with the known distribution pattern of PDE7 in the brain [100]. However, some activity accumulation was detected in the cerebral white matter as previously shown in rats and rhesus monkeys [99,100]. In these pre-clinical studies, the activity uptake in the cerebellum could not be reduced in blocking experiments and thus, the authors suggested that this might reflect non-specific binding of [^11^C]**25** rather than possible accumulation of brain-penetrating radiometabolites [100]. This assumption is further supported by the low inter-subject variability and the high time-stability of the estimated [^11^C]**25** V_T_ values in all brain regions over 60–80 min scan duration [100]. Metabolite analysis revealed around 50% of intact [^11^C]**25** in plasma at 90 min p.i. while two more polar radiometabolites were detected [100]. Accordingly, the in vivo degradation of [^11^C]**25** in humans seems to proceed slower than in monkeys (27% of intact radioligand at 90 min p.i. [99]). Overall, these preliminary results clearly demonstrate the suitability of [^11^C]**25** for imaging of the PDE7 enzyme in the human brain by PET, but further evaluation of the radioligand binding specificity and test-retest reproducibility of the [^11^C]**25** PET data in a larger cohort are required [100]. In conclusion, access to the first appropriate PDE7 radioligand might enable future (pre-)clinical PET studies including target occupancy assessments by PDE7 inhibitors and investigations of altered PDE7 availability in related brain disorders. 

## 7. PDE10A Radioligands

The PDE10A enzyme shows many similarities with the PDE2A isoform, namely the dual-substrate specificity by degrading both cyclic nucleotides cAMP and cGMP as well as the distribution pattern in the brain [3]. PDE10A is most abundantly expressed in the basal ganglia, predominantly in striatal medium spiny neurons, with highest levels in caudate nucleus and nucleus accumbens [101,102,103]. Accordingly and comparable to PDE2A, PDE10A activity is related to neurodegenerative and neuropsychiatric diseases like Huntington’s disease, Parkinson’s disease, Alzheimer’s disease, dementia, schizophrenia and depression [104,105,106,107,108,109].

### 7.1. Recent PET Studies Using Already Known PDE10A Radioligands

The well-known PDE10A-specific inhibitor **MP-10** [110] (see Figure 6) serves as molecular lead structure for various PDE10A radioligands as has already been reviewed [1,9]. In 2016, Ooms et al. [111] reported on in vitro and in vivo investigations of the correlation between intracellular cAMP levels and PDE10A activity using the structurally **MP-10**-related PDE10A radioligand [^18^F]**JNJ42259152** [112] ([^18^F]**26**, Figure 6). Binding studies on rat brain homogenates revealed a maximum specific binding of 0.75 nM and a *K*_D_ value of 6.62 nM for [^18^F]**26** at baseline conditions [111]. Increased cAMP concentrations significantly affected the *K*_D_ values in a dose-dependent manner with an up to 48% decrease at 10 µM cAMP indicating an increase of the in vitro PDE10A affinity of [^18^F]**26**. Ooms et al. suggested two possible reasons for that, namely the cAMP-induced (I) conformational change in the catalytic domain of PDE10A or (II) phosphorylation of PDE10A, both resulting in an elevated enzyme activity and thus, in an increased binding affinity of the radioligand [111].

To further investigate the cAMP-induced stimulation of PDE10A, PET studies in rats with [^18^F]**26** and inhibitors of PDE4 or PDE2A, to increase the cAMP levels in vivo, have been performed [111]. Besides PDE10A, the isoforms PDE4 and PDE2A are also expressed in the basal ganglia [3] as mentioned above in the respective paragraphs. The [^18^F]**26** BP_ND_ in striatum was significantly increased by 67% and 73% at 5 min and 60 min after pre-treatment with the PDE4 inhibitor **rolipram** (10 mg/kg). This result clearly shows that the higher [^18^F]**26** uptake was not induced by an elevated expression of PDE10A to normalize the cAMP levels which is expected to show only minimal effects at 5 min after **rolipram** treatment [111]. Pre-administration of the PDE2A inhibitor **JNJ49137530** (10 mg/kg; selectivity over PDE10A > 400 [111]) 30 min before [^18^F]**26** injection also revealed a 75% increase of the striatal BP_ND_ of the radioligand. This effect is suggested to be caused not only by higher cAMP levels since PDE2A is a dual-substrate specific enzyme like PDE10A and inhibition of PDE2A leads to a significant increase of cGMP as well. Hence, the observed higher striatal [^18^F]**26** binding might be induced by elevated concentrations of both cyclic nucleotides which needs to be further investigated [111]. In conclusion, this comprehensive study indicates an important role of cAMP on PDE10A activity as well as possible interactions between different PDEs in the brain [111]. These proposed mechanisms should be validated in future studies to prove whether prolonged treatment with PDE inhibitors also gradually activates the PDE10A enzyme and if this effect can be observed for other PDE isoforms.

In a 2017 published clinical study [113], [^18^F]**26** was used for assessment of the PDE10A availability in the striatum of patients with progressive supranuclear palsy (PSP) and PD to possibly differentiate between these two neurodegenerative disorders by PET. Compared to the healthy human brain, the striatal BP_ND_ of [^18^F]**26** was significantly lower in PSP patients and showed only slight decrease in PD patients, which is in contrast to previous results [45]. Koole et al. [113] suggested that the small cohort or the shorter disease duration of the PD group might cause this observation. The caudate nucleus-to-putamen BP_ND_ ratio of [^18^F]**26** was significantly reduced in both, patients with PSP and PD. Interestingly, there was no difference in the striatal BP_ND_ between the PSP and the PD group. Therefore, the authors concluded that PDE10A PET imaging might not be suitable for the often challenging clinical distinction between PSP and PD [113].

The next two structurally **MP-10**-related radioligands discussed here, [^18^F]**TZ19106B** and [^18^F]**TZ8110** [131] ([^18^F]**27** and [^18^F]**28**, see Figure 6), have already been evaluated in rats and monkeys showing their high potential as appropriate radioligands for PET imaging of PDE10A in the brain [1,119]. In 2018, further investigation of [^18^F]**27** and [^18^F]**28** was reported by Liu et al. [131] aiming at the quantitative comparison of their in vivo binding and imaging properties. PET studies in non-human primates revealed high uptake of both radioligands in striatum, fast clearance from non-target brain regions and good time stability of the striatal BP_ND_. However, [^18^F]**27** showed a considerably higher accumulation as well as retention in the striatum than [^18^F]**28** with SUV_max_ values of 1.76 at 90–100 min p.i. and 0.58 at 30–40 min p.i., respectively [131].

Hence, [^18^F]**27** was selected for displacement studies using **MP-10** (0.3–2 mg/kg) as blocking agent resulting in a significantly reduced striatal uptake with an up to 90% decreased BP_ND_ demonstrating a high PDE10A-specific binding. Furthermore, administration of **MP-10** (2 mg/kg) at 40 min p.i. of [^18^F]**27** revealed a considerable displacement of the radioligand in the striatum while activity levels in cerebellum were unaffected indicating reversible binding of [^18^F]**27**. PDE10A occupancy levels of about 35% at 0.3 mg/kg and 90% at 2 mg/kg of **MP-10** were estimated by [^18^F]**27** PET. Overall, these results confirm the high suitability of [^18^F]**27** for brain PET imaging and quantification of the PDE10A enzyme. Consequently, Liu et al. announced to apply for the United States Food and Drug Administration (FDA) approval for human application of [^18^F]**27** and to perform first clinical PET studies [131].

PDE10A plays a key role in the regulation of striatal signaling, which involve dopaminergic and cAMP-dependent pathways [132]. Particularly in medium spiny neurons, the cAMP levels are modulated by PDE10A and adenylyl cyclase (AC). PDE10A degrades cAMP while AC catalyzes the conversion of adenosine triphosphate to cAMP. Furthermore, AC activity is suppressed by dopamine D_2_ receptor (D_2_R) signaling. Liu et al. investigated the correlation between PDE10A and D_2_R in non-human primates using [^18^F]**27** PET and the D_2_R-selective inhibitor (-)-**eticlopride** [131]. Remarkably, acute pre-treatment with (-)-**eticlopride** (0.025 mg/kg) revealed a 34–44% increase of the striatal BP_ND_ of [^18^F]**27** [131]. This PDE10A upregulation is suggested to be induced by increased AC activity due to D_2_R inhibition and thus, elevated cAMP levels. Accordingly, quantification of PDE10A by PET might serve as indirect measure to investigate novel cAMP level modulating antipsychotic drugs [131].

In 2016, Lui et al. [133] reported on a comparative PET study of two PDE10A radioligands in cynomolgus monkeys, the **MP-10**-derived [^11^C]**TZ164B** [121,134] and the structurally not **MP-10**-related [^18^F]**MNI-659** [122,135,136,137] ([^11^C]**29** and [^18^F]**30**, see Figure 6). Both, [^11^C]**29** and [^18^F]**30**, have already been suggested as highly potent radioligands for in vivo imaging and quantification of PDE10A in the brain by PET based on pre-clinical ([^11^C]**29**) and first clinical ([^18^F]**30**) investigations [1]. For the head-to-head comparison of these two radioligands, double PET scans in the same monkey and at the same day have been performed [133]. [^11^C]**29** and [^18^F]**30** displayed a high accumulation in the striatum and an identical distribution pattern in the whole brain with lowest uptake in cerebellum [133]. For [^18^F]**30**, the averaged striatal BP_ND_ (5.1–5.3) and the area under the curve striatum-to-cerebellum ratio (AUC_St/Cb_ = 7.6) were higher than for [^11^C]**29** (BP_ND_ = 3.7–4.4; AUC_St/Cb_ = 5.9) but not statistically significant due to the large variation of the [^18^F]**30** PET data. These results correlated well with the higher in vitro binding affinity of [^18^F]**30** compared to [^11^C]**29** [133] (see Figure 6). Besides, [^11^C]**29** showed longer striatal retention (SUV_max_ ~ 2.7 at 40–60 min p.i.) and relatively slower washout from striatum than [^18^F]**30** (SUV_max_ ~ 4 at 10 min p.i.). Liu et al. discussed two possible reasons for that, first, the considerable smaller efflux rate constant of [^11^C]**29** (*k*_2_ = 0.13/min vs. 0.28/min for [^18^F]**30**) across the BBB. Second, the higher metabolic stability (70% of intact [^11^C]**29** vs. 50% of intact [^18^F]**30** in plasma at 60 min p.i.) and thus, a higher concentration of intact [^11^C]**29** in the brain [133]. The longer striatal [^11^C]**29** retention indicates a more stable binding of that radioligand to PDE10A which might improve the reproducibility of the obtained PET data especially at later time points [133]. In conclusion, the results of this comparative PET study additionally verify the suitability of [^11^C]**29** and [^18^F]**30** for in vivo assessment of the PDE10A enzyme in the brain. It appears that no further investigations of [^11^C]**29** have been reported since 2016. In contrast, numerous new insights have been published for [^18^F]**30** as will be discussed in the following paragraph. 

Regarding the radiosynthesis of [^18^F]**30**, Mori et al. [138] reported on the comparison between direct ^18^F-labeling of the corresponding tosylate precursor as previously described by Barret et al. [122] and a two-step approach by ^18^F-labeling of bromoethyl triflate and using the resulting 1-bromo-2-[^18^F]fluoroethane for ^18^F-fluoroethylation of the respective phenol precursor (see Scheme 12). The aim was to establish an appropriate strategy for routine production of the radioligand with high quality and sufficient activity for clinical application using an in-house developed automated multi-purpose synthesizer [138,139].

Mori et al. found out that the radiolabeling efficiency of the direct nucleophilic aliphatic substitution of the tosylate group using the K^+^/[^18^F]F^−^/K_2.2.2_-carbonate complex is highly dependent on the amounts of precursor and potassium carbonate [138]. A sufficiently high activity of [^18^F]**30** for clinical use (Scheme 12) was obtained by using at least 4 mg of the tosylate precursor and decreasing the amount of potassium carbonate to 2 nmol. Furthermore, the tosylate precursor proved to decompose under the reaction conditions, which has to be considered for separation of the radioligand from non-radioactive by-products by semi-preparative high-performance liquid chromatography (HPLC) [138]. In contrast, the ^18^F-fluoroethylation was performed with only 1.5 mg of the phenol precursor resulting in clinically useable doses of [^18^F]**30** (Scheme 12) while the radioligand could readily be purified by semi-preparative HPLC [138]. In conclusion, Mori et al. [138] have shown that both approaches, the direct ^18^F-labeling of the tosylate precursor and the two-step strategy by ^18^F-fluoroethylation of the phenol precursor, enable the automated radiosynthesis of [^18^F]**30** in high radiochemical and chemical purity as well as sufficient radiochemical yields and molar activities for human application.

As mentioned above, a functional association between pharmacological D_2_R inhibition and elevated PDE10A activity has been discovered by Liu et al. using [^18^F]**27** PET [131]. In 2017, Fazio et al. [140] reported on a related PET study aiming at (I) the quantitative age- and gender-related comparison of the availability of the dopamine D_2_ and D_3_ receptors (D_2/3_R) and the PDE10A enzyme in the healthy human brain, and (II) the assessment of the relative distribution of these two targets specifically in the striatum and globus pallidus. For these purposes, the D_2/3_R-dual specific radioligand [^11^C]**raclopride** and the PDE10A-selective radioligand [^18^F]**30** have been applied for brain PET imaging in a cohort of 40 participants with an age range between 27 and 69 years [140]. Regarding the relationship between D_2/3_R availability and age, only a trend of negative correlation was found by a decrease of 4.3% of the striatal [^11^C]**raclopride** binding per decade that is in line with previous findings. In the globus pallidus, no significant differences in D_2/3_R levels were observed between the age groups indicating rather stable receptor densities in this brain region consistent with former results. In contrast, the availability of the PDE10A enzyme in the striatum measured by [^18^F]**30** PET was significantly decreased by 8% per decade, as reported previously by Russel et al. [137], while a less pronounced decline was observed in the globus pallidus [140].

Interestingly, higher striatal BP_ND_ values were estimated for both radioligands in female participants compared to male subjects [140]. For [^18^F]**30**, this tendency was also observed in the globus pallidus. These results indicate gender-related effects on the availabilities of the D_2/3_R and especially of PDE10A, but further investigations in a lager cohort with a uniform distribution of male and female participants across the age groups are required. The regional distribution of [^11^C]**raclopride** displayed highest D_2/3_R density in the striatum (putamen > caudate nucleus) followed by the nucleus accumbens and the globus pallidus (external) consistent with previous outcomes [140]. [^18^F]**30** PET revealed high PDE10A availability in the striatum (caudate nucleus and putamen) and the globus pallidus (internal and external) and negligible enzyme levels in the nucleus accumbens. This distribution pattern is in agreement with immunohistochemical [101] and in vitro autoradiographic studies using [^18^F]**30** [141]. Besides, [^18^F]**30** BP_ND_ values showed high variability between individuals as already observed in previous PET studies in humans [122] and non-human primates [133]. Fazio et al. proposed that this might be caused either by the characteristics of [^18^F]**30** or by a possible inter-individual variability of PDE10A associated with different gene variants and activation states which might affect [^18^F]**30** binding to the enzyme. Overall, the regional uptake of [^11^C]**raclopride** and [^18^F]**30** in the human brain revealed a significant correlation of the D_2/3_R and PDE10A distribution in the striatum indicating a functional association between these two targets which is clearly supported by the above discussed findings of Liu et al. [133].

An [^18^F]**30** PET study for in vivo assessment of the PDE10A occupancy in the healthy human brain has been reported by Delnomdedieu et al. in 2017 [142]. The PDE10A inhibitor **MP-10** was used for that purpose. Additionally, this study was aiming to approve the ability of **MP-10** to enter the brain after oral administration as well as to estimate appropriate doses for future clinical trials by correlating its serum concentration and the percentage of enzyme occupancy. Application of 10 mg, 20 mg and 30 mg of **MP-10** revealed mean **MP-10** serum concentrations of about 31 ng/mL, 75 ng/mL and 99 ng/mL within 1.5 h after administration, respectively, indicating a dose-related increase of the systemic exposure to **MP-10** [142]. Briefly, a single oral dose of 30 mg of **MP-10** revealed adverse side effects in one volunteer, like fatigue, somnolence and musculoskeletal stiffness, and consequently lower doses were applied for further investigations [142]. Based on the [^18^F]**30** PET scans, PDE10A occupancies in whole striatum, caudate nucleus and putamen of around 21–28% at 10 mg and 52% at 20 mg of **MP-10** were estimated consistent with the observed increased systemic exposure of **MP-10** at higher doses. Notably, the mean occupancy values were consistent regardless whether or not the cerebellum was used as reference region for the calculations. The **MP-10** serum concentration associated with 50% PDE10A occupancy was estimated to be 93.2 ng/mL [142]. Overall, these results clearly confirm that **MP-10** reaches the striatum after oral administration and binds specifically to the PDE10A enzyme by displacing [^18^F]**30** in a dose-dependent manner. For future investigations of the pharmacological activity of **MP-10**, about 50% PDE10A occupancy at the well-tolerated single dose of 20 mg is considered sufficient. Moreover, the cerebellum is suggested to be suitable as a reference tissue in PDE10A-related PET data analysis. Finally, [^18^F]**30** proved to be an appropriate radioligand for reliable assessment of the PDE10A occupancy in the human brain by PET [142].

Furthermore, [^18^F]**30** PET has extensively been used for pre-clinical investigations in mouse models of Huntington’s disease (HD) [143,144] and for clinical studies in HD patients [1,136,137,145]. It is postulated that HD symptoms are caused by a dysfunction in the basal ganglia related to loss of PDE10A and reduced cyclic nucleotide signaling. Thus, PDE10A inhibitors are recommended as potent therapeutics of HD by elevating cAMP and cGMP levels, which leads to the compensation of basal ganglia circuitry deficits [143]. In 2016, Beaumont et al. [143] investigated the effects of acute PDE10A inhibition on regeneration of the basal ganglia circuitry in the Q175 and the R6/2 mouse models of HD that display reduced PDE10A expression comparable to manifest HD patients. For that purpose, the PDE10A-specific inhibitors **MP-10** and **PF-04898798** (*K*_D_(PDE10A) = 0.8 nM, >1000-fold selectivity over all other PDEs [143]) were used. The results of this comprehensive pharmacological study will not be discussed in detail here. In brief, the authors stated that despite extremely reduced PDE10A levels, acute inhibition of PDE10A in symptomatic HD mouse models results in considerable functional improvements, for example due to elevated cAMP and cGMP concentrations and enhanced striatal response to cortical stimulation [143]. Overall, these findings clearly provide rationale for application of PDE10A inhibitors in clinical trials to assess related symptom improvement in HD patients, which are currently in progress with **MP-10** [143].

In addition to these pharmacological studies, the striatal PDE10A availability was examined in Q175 and R6/2 mice compared to wild-type mice by autoradiography and PET. For ex vivo binding studies, the tritiated PDE10A radioligand [^3^H]**PF-04831704** (*K*_D_(PDE10A) = 0.066 nM in mouse striatum [146]) was used. A significant decrease of the striatal PDE10A *B*_max_ was found in both HD mouse models, in R6/2 mice at early- and late-symptomatic stages (6 and 15 weeks) while in Q175 mice this reduction was only significant at mid- and late-symptomatic stages (8 and 12 month). The stage-related decline of PDE10A levels in the striatum is consistent with the results from PET studies using [^18^F]**30** in Q175 mice at the mid-symptomatic stage (6 month), where a significant decrease of the striatal [^18^F]**30** BP_ND_ of 52% was found. Moreover, [^11^C]**raclopride** PET revealed a reduction of 40% of striatal D_2/3_R levels in Q175 mice at the same disease stage. These results indicate a correlation between PDE10A and D_2/3_R in HD additionally to the above discussed findings of Fazio et al. [140] and Liu et al. [131] under physiological conditions.

In 2018, Bertoglio et al. [144] investigated the accuracy of [^18^F]**30** PET to image altered PDE10A enzyme levels in the Q175 mouse model of HD at the mid-symptomatic stage (6 month) compared to wild-type mice. Regional quantification and spatial normalization of the respective PET images were performed based on additional X-ray computed tomography (CT) and magnetic resonance imaging (MRI) or the [^18^F]**30** PET data only. Individual MRI scans revealed a significant reduction of the striatal volume of 7.7% in Q175 mice consistent with previous findings in animal models of HD [144]. The estimated striatal BP_ND_ of [^18^F]**30** was decreased by 47%, 43% and 32% using the CT, MRI or PET template-based spatial normalization, respectively. However, the mean [^18^F]**30** BP_ND_ in the striatum of Q175 mice based on the PET template was significantly higher than for CT- or MRI-related normalizations indicating an overestimation of the PDE10A availability in the PET template-based approach [144]. Regarding the CT template, larger variability of the calculated BP_ND_ values was observed resulting in reduced statistical power of these data. Thus, the authors stated that the PET and CT templates display lower detectability of the HD-related decline of the PDE10A enzyme density in the mouse brain [144]. Furthermore, striatal volume delineated manually from individual MRI scans was used as additional independent measurement to quantify [^18^F]**30** BP_ND_. Comparison of these estimated BP_ND_ values with the normalized BP_ND_ values from the CT, MRI and PET templates revealed highest correlation for the use of the MRI template. In conclusion, MRI-based spatial normalization is highly recommended for brain PET imaging using [^18^F]**30** to accomplish correct PDE10A quantification and improve the detectability of HD-related effects [144].

In a recently published clinical PET study, Fazio et al. [145] reported on the evaluation of PDE10A and D_2/3_R imaging as biomarkers of the progression of HD using [^18^F]**30** and [^11^C]**raclopride**. This comprehensive study included 44 patients with early and late pre-manifest HD gene-expansion carriers (HDGECs) as well as manifest HDGECs of stage 1 and 2. Compared to healthy controls, [^18^F]**30** PET exhibited significant decreases of the mean BP_ND_ of 41% in the caudate nucleus, 42% in the putamen and 38% in the globus pallidus averaged for all HDGECs groups reflecting a considerable loss of PDE10A in these brain regions [145]. Regarding D_2/3_R levels, a similar trend was observed in the caudate nucleus and putamen with an averaged decrease of the [^11^C]**raclopride** BP_ND_ values of 32% in all HDGECs groups while no significant changes were detected for the globus pallidus due to the low D_2/3_R expression in this basal ganglia region [145]. Within the different HDGECs groups, the striatal PDE10A availability seems to be characterized by a progressive decline between early and late pre-manifest stages (29–36%), as well as between the late pre-manifest stage and manifest stage 1 (39–52%). The same significant stage-related decline was observed for D_2/3_R levels in the caudate nucleus and putamen but the decrease of PDE10A occurred earlier and was more pronounced [145]. Additionally, longitudinal studies at around 22 months after the first PET scans were performed to assess the effect of the respective disease stage on the rates of change in the PDE10A availability. These data revealed a further decrease of PDE10A over time with higher annualized rates of change in the caudate nucleus (5.9%) than in the putamen (4.4%) and the globus pallidus (4.3%) while a trend of faster decline in the earliest disease stages was observed [145]. Overall, these findings indicate a potential role of PDE10A as neuroimaging biomarker of HD to evaluate phenoconversion at early disease stages as well as pharmacodynamic effects of disease-modification strategies during pre-symptomatic phases [145].

Besides the relation between altered PDE10A availability in the basal ganglia and the progression of HD discussed above, previous findings provide evidence that changes of the PDE10A expression in extra-striatal brain regions might also be an important pathophysiological feature in that regard [147,148,149]. In 2016, Wilson et al. [150] reported on a clinical PET study to examine quantifiable extra-striatal PDE10A levels in the healthy human brain and in patients with early pre-manifest HDGECs. For that purpose, the PDE10A-specific radioligand [^11^C]**IMA107** [116] ([^11^C]**31**, see Figure 6) was used. The suitability of [^11^C]**31** for in vivo imaging and quantification of PDE10A in the human brain by PET has already been proven [1,45,116,151,152]. Notably, MRI scans revealed no differences in the volumes of any extra-striatal brain region between healthy controls and the early pre-manifest HDGECs group indicating no effect of regional atrophy on the intracellular PDE10A levels in these brain areas [150]. For accurate detection of the relatively low extra-striatal PDE10A levels [101] by [^11^C]**31** PET, a BP_ND_ value of over 0.3 in healthy controls was set as threshold criterion [150]. In the early pre-manifest HDGECs group, the [^11^C]**31** BP_ND_ was significantly decreased by 25% in the insular cortex and by 42% in the occipital fusiform gyrus demonstrating a considerable loss of PDE10A in these brain tissues [150]. This [^11^C]**31** BP_ND_ decline correlated well with the reduced radioligand accumulation in the target region striatum indicating that the activity signal in the PDE10A-poor brain areas is caused by specific uptake of [^11^C]**31** instead of background noise [150]. In conclusion, the findings of Wilson et al. support that dysregulation of PDE10A in HD (I) is a very early pathophysiological event in the disease progression, and (II) does not only occur in the striatum but also in the insular cortex and the occipital fusiform gyrus. Loss of PDE10A in these extra-striatal brain regions might be associated with the risk of developing cognitive and behavioral disorders in manifest HD, but this has to be further investigated [150].

In 2016, Diggle et al. [153] reported on the investigation of *PDE10A* gene mutations associated with hyperkinetic movement disorders (hMDs). In eight participants affected by an early-onset hMD, genetic mapping and whole-exome sequencing identified the homozygous *PDE10A* mutations p.Tyr107Cys and p.Ala116Pro [153]. [^11^C]**31** PET was performed to quantify the striatal PDE10A levels in one person with the p.Tyr107Cys variant compared to healthy controls. In this hMD participant, the [^11^C]**31** BP_ND_ values were significantly decreased by 70% in the caudate nucleus, putamen and globus pallidus indicating substantial reduction of the PDE10A availability in these basal ganglia regions. MRI scans revealed no differences in striatal volumes of participants with homozygous *PDE10A* mutations and healthy controls confirming that the decreased PDE10A levels observed by [^11^C]**31** PET are not caused by regional atrophy but by striatal dysfunction. Additionally, PDE10A availability was examined in a motor phenotype displaying knock-in (KI) mouse model with the p.Tyr97Cys-mutated variant of *PDE10A*, which is homologous to the human p.Tyr107Cys variant. Immunoblotting and cAMP-related enzyme activity assays revealed significantly reduced PDE10A expression and activity in the KI mouse striatum compared to wild-type mice consistent with the results from the human PET study. Furthermore, administration of the PDE10A inhibitor **MP-10** in wild-type mice exhibited an up to 400% increased striatal level of the phosphorylated cAMP-response element binding protein (pCREB), which represents an indirect measure of the intracellular cAMP concentration. In contrast, no significant changes in the striatal pCREB levels were observed in KI mice after **MP-10** treatment indicating a tremendously reduced downstream pCREB signaling in the mutated PDE10A variant. Although the authors provided no information about the applied **MP-10** doses in this study, it would be of high interest whether increased concentrations of **MP-10** could induce a pCREB response in the KI mouse model. Moreover, it should be considered whether the inhibitory potency (IC_50_) or binding affinity (*K*_D_) of [^11^C]**31** and **MP-10** towards the mutated PDE10A enzymes differ from the wild-type form, which could be of high impact for related imaging and pharmacological studies. In conclusion, Diggle et al. proposed that homozygous mutations in the *PDE10A* gene cause extensive loss of the PDE10A enzyme in the striatum resulting in striatal dysfunction and, thus, leading to hMDs [153]. 

In another related study, Niccolini et al. [154] investigated the integrity of striatocortical pathways in two genetic hMDs associated with mutations of PDE10A and adenylyl cyclase 5 (AC5). PDE10A and AC5 regulate the intracellular cAMP levels in the striatal medium spiny neurons as mentioned before. Mutations in the *PDE10A* and *AC5* genes result in functional dysregulation of these enzymes and increased cAMP levels, which is suggested to cause hMDs [154]. The study by Niccolini et al. included six hMD participants with heterozygous mutations in *PDE10A* (p.Phe300Leu and p.Ile625Phe/p.Glu67Gln) or *AC5* (p.R418W) and 16 healthy controls [154]. Quantification of the striatal PDE10A availability was assessed by [^11^C]**31** PET. In two participants with the p.Phe300Leu variant of *PDE10A*, PET imaging revealed a significantly decreased BP_ND_ of [^11^C]**31** in the caudate nucleus (54% or 76%), putamen (71% or 82%) and globus pallidus (66% or 79%) demonstrating considerably reduced PDE10A availability [154]. Interestingly, the participant with the p.Ile625Phe/p.Glu67Gln variant of *PDE10A* displayed a much lesser decline of the PDE10A levels (14–59%) [154]. Moreover, PET scans in two participants with the *AC5* mutation p.R418W revealed a significant decrease of 47% or 67% of the [^11^C]**31** BP_ND_ in the globus pallidus while no changes in the striatum were detected [154].

Additionally, all participants underwent single-photon emission computed tomography (SPECT) using [^123^I]**FP-CIT**, which is a selective radioligand for quantification of the dopamine transporter (DAT). In the subjects with the p.Phe300Leu variant of *PDE10A*, SPECT scans displayed a significant decrease of 40% or 54% of the [^123^I]**FP-CIT** specific binding ratio (SBR) in the caudate nucleus. Similarly, participants with the *AC5* mutation p.R418W showed a reduction of 32% or 57% of the [^123^I]**FP-CIT** SBR exclusively in the caudate nucleus. These results indicate an impaired striatal pre-synaptic dopaminergic terminal integrity [154].

Furthermore, MRI scans of all participants were performed for volumetric analysis of the related brain regions. No significant atrophy was found in the striatum of participants with the p.Ile625Phe/p.Glu67Gln variant of *PDE10A* or the *AC5* mutation p.R418W. In the two participants with the *PDE10A* mutation p.Phe300Leu, significant volume loss in the caudate nucleus (31% or 45%), putamen (32% or 48%) and globus pallidus (41% or 50%) was observed. This outcome indicates that the decreased PDE10A and DAT levels detected by PET and SPECT might be caused by both, basal ganglia dysfunction as well as atrophy due to the p.Phe300Leu mutation in the *PDE10A* gene. Thus, the heterozygous p.Phe300Leu mutation might induce different pathophysiological mechanisms leading to a different clinical phenotype compared to homozygous *PDE10A* mutations as investigated by Diggle et al. [153] and discussed above. Overall, these results point to an association between *PDE10A* and *AC5* mutations and a pathological reduced expression of PDE10A and DAT in the basal ganglia as well as dysfunctions within the striatocortical pathways leading to hMDs [154].

In that regard, Pagano et al. [155] further investigated the relationship between altered expressions of PDE10A and DAT and the progression of motor symptoms, especially in early stages of PD. In a cohort of 17 early de novo and 15 early-L-DOPA-treated PD patients as well as 22 healthy controls, brain PET imaging was performed using [^11^C]**31** and the DAT-specific radioligand [^11^C]**PE2I** [155]. The BP_ND_ of both [^11^C]**31** and [^11^C]**PE2I** was significantly decreased in the whole striatum, caudate nucleus, putamen and ventral striatum of early de novo PD patients with less than two years of disease duration. In the globus pallidus (internal and external) and substantia nigra, only reduced binding of [^11^C]**PE2I** was observed [155]. Moreover, no differences between the [^11^C]**31** BP_ND_ values in contralateral brain regions of early de novo PD patients with unilateral motor symptoms were detected demonstrating a bilateral reduction of PDE10A. In contrast, [^11^C]**PE2I** binding was decreased in brain areas related to the clinically most affected side of the body displaying that loss of DAT was only unilateral [155]. Early-L-DOPA-treated PD patients, which had a three years longer disease duration than the early de novo PD subjects, showed an additional reduction of the [^11^C]**31** BP_ND_ of 17% and 10% in the caudate nucleus and putamen, respectively. For [^11^C]**PE12I**, further decreased binding of 35% was exclusively detected in the putamen [155]. These results clearly point to a reduced availability of both PDE10A and DAT in early stages of PD. Compared to healthy controls, PET imaging in the PD groups revealed a greater decrease of the PDE10A levels in the caudate nucleus while in the putamen, globus pallidus and substantia nigra the loss of DAT was more pronounced [155]. Notably, MRI exhibited no volumetric differences in the respective striatal tissues or the globus pallidus between PD groups and healthy controls indicating that the reduced binding of [^11^C]**31** and [^11^C]**PE2I** was not caused by regional atrophy. Overall, Pagano et al. concluded that there is an association between second messenger signaling and dopaminergic function and that pathological dysregulation of postsynaptic PDE10A might occur prior to dopaminergic terminal loss in the early progression of motor symptoms in PD [155].

Besides the application of [^11^C]**31** PET for the investigation of altered PDE10A availability in movement disorders, Tollefson et al. [156] used this approach to examine whether chronic cocaine use induces changes of the PDE10A levels in the human brain. Briefly, previous studies in rodents have shown an increased density of striatal medium spiny neurons (MSN) following exposure to cocaine, which has been reported to persist, even after 14–30 days of abstinence [156,157,158,159]. Therefore, and due to the high expression of PDE10A in MSN, Tollefson et al. assumed that participants affected by cocaine use disorder might display elevated PDE10A levels in the striatum [156]. However, PET imaging in 15 cocaine-consuming participants that were abstinent for at least 10 days revealed no significant differences in the [^11^C]**31** BP_ND_ in striatal regions, midbrain, thalamus or globus pallidus compared to healthy controls [156]. A negative trend was observed in the caudate nucleus and globus pallidus with a decreased binding of [^11^C]**31** of 15% and 9%, respectively. Notably, MRI scans displayed a volumetric loss of 10% in the caudate nucleus of participants with cocaine use disorder, which was unexpected and not consistent with previous findings [156]. After partial volume correction of the [^11^C]**31** BP_ND_ values, a slight decrease of the radioligand binding of about 8% in the caudate nucleus and 9% in the globus pallidus was still detected but not significant. Regarding statistical power of these data, Tollefson et al. suggested that PET studies in a larger cohort are required. Additionally, the authors stated further possible reasons for the negative results obtained, e.g., species differences that might cause the inconsistency of the human PET data with the findings in rodents after cocaine administration. Besides, the PDE10A levels might not only be influenced by altered MSN density, but also by cocaine-induced alterations in the dopamine and glutamate transmission, which affect the cyclic nucleotide signaling [156]. Finally, Tollefson et al. concluded that the negative outcome of this imaging study further diminishes the prospects for PDE10A inhibitors as a therapeutic target in cocaine use disorder [156].

In 2017, Yang et al. [160] reported on the in vivo evaluation of another already known PDE10A-specific radioligand, [^11^C]**LuAE92686** [124,125,126] ([^11^C]**32**, see Figure 6), in non-human primates. The suitability of [^11^C]**32** for PET imaging and quantification of PDE10A in the target region striatum has previously been proven in non-human primates and humans [1,124]. The aim of the current PET study was to further characterize the binding of [^11^C]**32** to PDE10A regarding specificity and kinetics and to examine whether [^11^C]**32** is suitable to accurately quantify PDE10A levels in the substantia nigra where this enzyme is exclusively expressed in nerve fibers and terminals [101,160]. The substantia nigra is a key nucleus in relation to the basal ganglia and thus, regional dysfunction of PDE10A is suggested to be associated with the pathophysiology of neuropsychiatric disorders like schizophrenia [160]. In that regard, correct quantification of PDE10A in this brain tissue by PET is needed to investigate the effect of altered enzyme density or activity. PET scans at baseline conditions revealed high binding of [^11^C]**32** in the target region striatum (BP_ND_ ~ 12), as expected, and mean BP_ND_ and V_T_ values of 3.7 and 2.3 in the substantia nigra [160]. Notably, binding to PDE10A in the substantia nigra has been reported for two other radioligands, [^18^F]**26** and [^11^C]**31** (Figure 6), with a BP_ND_ of only 0.4 and 0.5, respectively. Yang et al. suggested that this might be caused by lower resolution PET systems in these previous studies [160]. This assumption is further supported by the fact that the striatal BP_ND_ of [^11^C]**32** was 85% higher compared to earlier findings in non-human primates due to the use of a high resolution research tomograph (1.6 mm vs. 3.5 mm) and a head fixation device in the present study [124,160]. Pre-treatment with either **MP-10** (1.5 mg/kg) or **32** (0.5 and 2 mg/kg) displayed significantly reduced uptake of [^11^C]**32** in all target regions confirming specific binding of the radioligand. In the substantia nigra, a decrease in the [^11^C]**32** V_T_/fp of around 45% using **MP-10** and up to 76% for **32** as blocking agents was observed. The estimated PDE10A occupancies were about 94% for **MP-10** as well as 84% and 96% for the two doses of **32**, respectively [160].

Metabolism studies revealed a fast degradation of [^11^C]**32** with only 18% of intact radioligand in plasma at 60 min p.i. [160] consistent with former results [1,124]. Several more polar radiometabolites were formed and it could not be excluded that part of these are able to enter the brain as observed in rats (<15% of radiometabolites in the brain at 40 min p.i.) [124,160]. Interestingly, the V_T_ values in the reference region cerebellum continuously increased during the PET scans from 63 min to 123 min p.i. [160] indicating that activity quantification in this non-target area might be influenced by the presence of brain-penetrating radiometabolites. Consequently, Yang et al. proposed to shorten the PET imaging duration to 63 min when using [^11^C]**32**, which is sufficient for reliable estimation of V_T_ values in the non-human primate brain and minimizes the potential contribution from radiometabolites [160]. However, identification of the brain-penetrating radiometabolites and investigation of their passage across the BBB are needed to clarify their impact on the quantification of [^11^C]**32** binding [160]. Notably, a considerably higher in vivo stability of [^11^C]**32** in humans has been reported (~70% of intact radioligand in plasma at 60 min p.i. [124]) and thus, the relevance of brain-penetrating radiometabolites discussed here might be negligible for clinical studies [160]. Overall, these outcomes clearly support the suitability of [^11^C]**32** PET for quantification of PDE10A levels in striatal brain regions and the substantia nigra due to the high signal-to-noise ratio and the specific binding of the radioligand [160].

Also in 2017, Bodén et al. [161] published the first clinical [^11^C]**32** PET study in patients with schizophrenia. The aim of this study was to assess whether there is a link between striatal dysfunction and frequently observed cortical thinning that is already present at early stages of schizophrenia [161]. For that purpose, quantification of striatal PDE10A levels by PET using [^11^C]**32** was performed in 16 healthy controls and 10 schizophrenia patients treated with diazepines. Additionally, all participants underwent MRI scans for measurement of cortical thickness [161]. PET scans revealed significantly reduced [^11^C]**32** BP_ND_ values in the striatum (caudate nucleus and putamen) of patients with schizophrenia while no differences were observed in the globus pallidus and substantia nigra compared to healthy controls [161]. These results indicate a decreased availability of striatal PDE10A in schizophrenia. Bodén et al. discussed that alterations in PDE10A levels might be caused either by the pathophysiology of the disease or by chronic antipsychotic treatment due to a possible direct or indirect inhibition of PDE10A [161]. By the latter, the observed reduction of the PDE10A availability would only reflect a treatment effect. This is not expected, because it is more likely that diazepines activate the dopamine D_1_ receptor (D_1_R) [162] which would lead to elevated cAMP concentrations and thus, to increased PDE10A activity or density [161].

Moreover, a positive correlation between the striatal [^11^C]**32** BP_ND_ and cortical thickness in the left superior frontal gyrus and the anterior cingulate cortex was observed in all participants. This outcome indicates an important role of PDE10A in striato-cortical interactions and that striatal dysfunction and cortical thinning are part of the same underlying pathophysiology in schizophrenia [161]. However, additional studies are required to clarify whether reduced PDE10A availability is primary or secondary to cortical thinning [161]. In summary, Bodén et al. concluded that the observed lower striatal PDE10A expression in patients with schizophrenia has to be verified, preferably in longitudinal studies of treated and untreated patients with early disease stages to elucidate whether this is a pathophysiological feature or an effect of antipsychotic treatment [161].

In 2019, Persson et al. [163] published a combined [^11^C]**32** PET and functional MRI (fMRI) follow-up study in the same cohort of schizophrenia patients like Bodén at al. [161]. This study focused on the investigation of the effect of reduced PDE10A levels in the striatum on the neuronal and behavioral function of striatal and downstream basal ganglia regions [163]. FMRI was used to measure striatal function and activity, the latter as the amplitude of low-frequency fluctuations (ALFF) [163]. PET/fMRI scans revealed a significant correlation between decreased [^11^C]**32** BP_ND_ and increased ALFF exclusively in the putamen and the substantia nigra [163]. These results indicate a reduced PDE10A availability and an elevated excitability in the respective brain regions, consistent with previous findings where a hyperactivity in the putamen of schizophrenia patients was observed [163,164,165,166]. The authors suggested that the higher excitability is caused by increased levels of cAMP and cGMP resulting from the loss of PDE10A [163,167]. The increased activity in the substantia nigra is supposed to reflect a downstream effect of decreased PDE10A levels in the striatum [163]. Furthermore, the ALFF in the putamen and substantia nigra significantly correlated with electrophysical and behavioral measures indicating that higher ALFF are associated with lower performance and thus, cognitive impairments in schizophrenia [163]. Regarding the possible indirect effect of diazepine treatment on altered PDE10A availability discussed by Bodén et al. [161], there was no relation found between the reduced striatal PDE10A levels and antipsychotic dosage in the present study. This is reflected by the fact that the ALFF in the putamen and substantia nigra did not correlate with the olanzapine equivalent dosage [163]. However, the complex functional link of the dopaminergic and striatal pathways needs to be further investigated. In conclusion, this follow-up study provided evidence for a relation between decreased PDE10A levels in the striatum and basal ganglia function, both at the neural and behavioral level, in schizophrenia [163].

In 2016, another already known PDE10A-specific radioligand, [^11^C]**T-773** [127,128,168] ([^11^C]**33**, see Figure 6), was evaluated in a pilot in-human PET study by Takano et al. [169]. Briefly, the favorable imaging profile of [^11^C]**33** regarding brain kinetics, specificity of binding and dosimetry has previously been demonstrated in non-human primates [1,127,128,170,171]. However, blocking studies have shown contrary results due to an unexpected decrease of the [^11^C]**33** V_T_ in the non-target regions cerebellum and cortex after pre-treatment with the PDE10A-specific inhibitor **TAK-063** [129] (see Figure 6) but not with **MP-10** [1,170,171]. The aim of the first clinical [^11^C]**33** PET study was to establish an appropriate method for correct quantification of the radioligand uptake by evaluation of its distribution in the human brain, the test-retest variability of the PET data as well as the effect of **TAK-063** in blocking and occupancy experiments [169]. PET scans revealed rapid uptake of [^11^C]**33** in the brain that peaked within 7.5 min, highest accumulation in the striatum with V_T_ values of 4.4 and 5.5 for the caudate nucleus and putamen, respectively, and a fast washout [169]. Furthermore, good reproducibility of the [^11^C]**33** V_T_ within 123 min scan duration and high time stability in all brain regions with 57 min shortened PET data were observed [169]. Metabolite analysis exhibited about 64% and 52% of intact [^11^C]**33** in plasma at 30 min and 75 min p.i., respectively, while more polar radiometabolites were detected, consistent with the in vivo degradation of the radioligand in non-human primates [127,128,169,170]. Administration of a single oral dose of **TAK-063** (3–1000 mg) at 3 h prior to the radioligand injection resulted in a significantly decreased V_T_ of [^11^C]**33** in the striatum (37–68% in putamen, dose-dependent), as expected, and in the non-target regions cerebellum (around 64%, independent from the dose), cortex, thalamus and hippocampus [169]. In all non-target brain areas, a similar blocking effect of **TAK-063** was observed, which is in line with previous findings in non-human primates [171]. These results clearly point to unknown off-target interactions of [^11^C]**33** and **TAK-063** due to their same molecular core structure (Figure 6) and the negligible PDE10A levels in the respective brain tissues [169,171]. Thus, Takano et al. have previously reported on a novel approach for the estimation of target occupancies using **TAK-063** by correcting the PET data for cerebral binding of [^11^C]**33** [169,171]. In the present study, the PDE10A occupancy obtained showed a dose-dependent increase from 3% to 72% at 3 mg and 1000 mg of **TAK-063**, respectively, which correlated with elevated plasma concentrations of the blocking agent [169]. Nevertheless, Takano et al. suggested that a comparative study using **MP-10** for the assessment of the PDE10A occupancy in the human brain might be valuable to understand these outcomes [169]. In conclusion, this first human [^11^C]**33** PET study further confirmed the promising imaging characteristics of the radioligand in terms of in vivo quantification of PDE10A in the brain and reproducibility of the PET data [169].

Besides the complex role of PDE10A in the pathophysiology of several brain disorders discussed above, there is strong evidence that PDE10A is involved in the regulation of whole body energy balance and thus, selective PDE10A inhibitors are proposed as novel therapeutics for the treatment of obesity [1,172,173]. We have previously demonstrated a significant upregulation of PDE10A in the brown adipose tissue (BAT) of various mouse models of obesity by PET/MRI using our PDE10A-specific radioligand [^18^F]**AQ28A** [130,173,174] ([^18^F]**34**, see Figure 6). In a follow-up study [175,176], we aimed to investigate further the relation between PDE10A activity and the regulation of energy homeostasis. For that purpose, [^18^F]**34** PET studies were performed in normal weight (NW), diet-induced obese (DIO) and genetically obese (ob/ob, leptin deficient) mice [175,176]. In NW mice, an intense symmetric [^18^F]**34** uptake in the interscapular BAT was observed, which was significantly higher compared to the surrounding skeletal muscle at 20–30 min p.i. (SUV = 0.55–0.6 vs. 0.3–0.4) [175]. Notably, blocking with **MP-10** revealed a substantially reduced accumulation of [^18^F]**34** in the BAT confirming specific binding of the radioligand and thus, demonstrating a marked expression of PDE10A consistent with real-time quantitative polymerase chain reaction (PCR) analysis of mouse and human BAT depots [175]. In DIO and ob/ob mice, the [^18^F]**34** uptake was significantly increased by about 80% in the BAT and 25–75% in the striatum compared to NW mice indicating a noticeably elevated PDE10A availability in obesity [175,176].

Additionally, [^18^F]**FDG** PET was performed to investigate the effect of PDE10A activity on the glucose metabolism [175,176]. Acute pharmacological inhibition of PDE10A using **MP-10** (30 mg/kg) revealed a significantly higher [^18^F]**FDG** uptake in the BAT of NW mice (SUV = 0.40 at 55 min p.i.) but not of DIO mice compared to vehicle groups (SUV = 0.25 at 55 min p.i.) [175,176]. Besides, administration of **MP-10** in NW mice protected against hypothermia during a four hours cold challenge at 8 °C while accumulation of [^18^F]**FDG** in the BAT was significantly reduced [175]. Overall, these results suggest a predominantly peripheral effect of **MP-10**-mediated inhibition of PDE10A activity on the metabolic function in the BAT associated with thermogenesis [175,176]. Notably, chronic administration of **MP-10** (10 mg/kg per day for one week) to DIO mice resulted in a significant weight loss of around 9% and an improved insulin sensitivity compared to the vehicle group [175]. Moreover, **MP-10** treatment of NW mouse adipose tissue depots led to increased cyclic nucleotide concentrations and reduced PDE10A expression in the BAT and the visceral white adipose tissue (VAT), as well as browning of the VAT as also shown in human adipose tissue depots [175].

Furthermore, a retrospective analysis of whole-body PET images from previous human dosimetry studies using [^18^F]**30** (Figure 6) [122] displayed a noticeable uptake of the radioligand in the supraclavicular BAT, which was significantly higher in individuals with a higher body mass index (BMI) [175]. These observations together with the time quantitative PCR analysis of adipose tissue depots demonstrate for the first time that PDE10A is expressed in the human BAT and might be upregulated in obesity. However, further studies in a larger cohort of participants are required to validate this outcome and draw definitive conclusions. In summary, this follow-up study demonstrated that PDE10A is an important regulator of at least three major processes in the BAT, namely thermogenic gene expression, lipolysis, and glucose uptake [175]. In addition, our results further provide evidence that pharmacological inhibition of PDE10A might be a promising strategy for the treatment of obesity and diabetes.

### 7.2. Novel PDE10A Radioligands

In 2018, Stepanov et al. [177] reported on the development of two novel ^18^F-fluoroalkylated analogs of [^11^C]**33** ([^11^C]**T-773**, Figure 6), namely [^18^F]**FM-T-773-d_2_** and [^18^F]**FE-T-773-d_4_** ([^18^F]**35** and [^18^F]**36**, see Figure 7). Briefly, [^11^C]**33** has already been evaluated in humans and presented an appropriate imaging profile for in vivo assessment of the PDE10A availability in the brain by PET although unknown off-target interactions of [^11^C]**33** under blocking with the structurally related PDE10A-specific inhibitor **TAK-063** (Figure 6) have been observed [169] as discussed above. Thus, Stepanov et al. selected **33** as lead compound and replaced the *O*-methyl group at the pyridazine core with deuterated fluoromethyl or fluoroethyl chains [177]. The novel ligands **35** and **36** displayed slightly lower but still high potency and selectivity towards the PDE10A enzyme compared to **33** (Figure 6 and Figure 7) and were proposed as suitable PET radioligands.

The radiosyntheses of [^18^F]**35** and [^18^F]**36** were performed in a two-step approach via ^18^F-labeling of 1,2-dibromomethane-d_2_ or 2-bromoethyl-d_4_ 4-methylbenzenesulfonate followed by reaction of the resulting [^18^F]fluoroalkyl synthons with the respective desmethyl precursor [177] (see Scheme 13).

Baseline PET studies in non-human primates revealed high and heterogeneous accumulation of both radioligands in the brain [177]. The uptake of [^18^F]**35** and [^18^F]**36** was characterized by peak values of 5.5% ID and 3.5% ID as well as half-lives of 20 min and 30 min, respectively. The binding of [^18^F]**35** and [^18^F]**36** specifically in the putamen reached equilibrium within 30 min and 45 min while putamen-to-cerebellum ratios of about 3 for [^18^F]**35** and less than 2 for [^18^F]**36** were observed [177]. Metabolism studies displayed moderate in vivo stability of both radioligands with 71% and 58% of intact [^18^F]**35** and [^18^F]**36** in plasma at 90 min p.i., respectively, while no defluorination was observed [177]. Particularly [^18^F]**35** seems to be rather more stable than [^11^C]**33** in non-human primates and humans (45% and 52% of intact radioligand at 90 min or 75 min p.i. [128,169]). Except for one lipophilic radiometabolite of [^18^F]**35** that was detected in only negligible amounts, mainly more polar radiometabolites of [^18^F]**35** and [^18^F]**36** were formed [177], consistent with the in vivo degradation of [^11^C]**33** [127,128,169,170,171]. Based on these results, Stepanov et al. concluded that [^18^F]**35** showed superior imaging features over [^18^F]**36** due to the higher brain uptake and binding to striatal regions as well as the favorable kinetics [177].

Consequently, [^18^F]**35** was selected for further investigations including blocking and occupancy studies using the PDE10A inhibitor **MP-10** (Figure 6) [177]. Pre-administration of **MP-10** (1.8 mg/kg) revealed significant decreases of the V_T_ and BP_ND_ of [^18^F]**35** in the caudate nucleus (15% and 68%) and putamen (23% and 64%) while no blocking effect in the cerebellum, thalamus and cortex was observed indicating specific binding of the radioligand [177]. However, it would be of interest to know whether or not pre-administration of **TAK-063** might cause a reduced binding of [^18^F]**35** in the non-target brain regions due to off-target interactions as shown previously for the structurally related [^11^C]**33** [169,171]. A PDE10A occupancy of about 60% in the caudate nucleus and putamen was estimated for **MP-10** (1.8 mg/kg) by [^18^F]**35** PET [177]. In conclusion, [^18^F]**35** displayed a promising imaging profile that is highly comparable with [^11^C]**33** and thus, further evaluation of this novel PDE10A radioligand in clinical PET studies is planned [177].

In 2019, four novel ^18^F-labeled radioligands, [^18^F]**37–40** (see Figure 7), for PET imaging of the PDE10A enzyme in the brain were published by Mori et al. [123]. This study focused on the development of a more suitable PDE10A radioligand than the above discussed and clinically evaluated [^18^F]**30** [122,136,137,140,142,145] ([^18^F]**MNI-659**, Figure 6) regarding its in vivo degradation and the possible formation of a brain-penetrating radiometabolite [123]. Notably, detailed pre-clinical data of the metabolite analysis of [^18^F]**30** in the brain have not been reported so far [123]. Based on the molecular structure of **30**, Mori et al. generated four novel PDE10A-affine analogs with different fluoroalkoxy side chains (**37–40**, Figure 7) and examined the in vivo profiles of the respective ^18^F-labeled radioligands in rodents directly compared with [^18^F]**30** [123]. The two-step radiosyntheses of [^18^F]**37–40** were performed by preparation of the corresponding [^18^F]fluoroalkylating agents and subsequent reaction with the phenol precursor in a home-made automated synthesis system [123,139] (see Scheme 14).

Baseline PET scans with [^18^F]**37**, [^18^F]**39** and [^18^F]**40** in rats revealed high initial brain uptake, a heterogeneous distribution and strongest accumulation in the striatum with BP_ND_ values of 3.8, 5.6 and 5.8, respectively [123]. In contrast, the striatal uptake of [^18^F]**38** was negligible (BP_ND_ = 0.6) indicating a limited penetration of this radioligand through the BBB [123]. Compared to [^18^F]**30** with a striatal BP_ND_ of 4.6, the radioligands [^18^F]**39** and [^18^F]**40** displayed a significantly higher binding to the striatum and thus, were further investigated [123].

Biodistribution studies in mice exhibited similar patterns of [^18^F]**39**, [^18^F]**40** and [^18^F]**30** throughout the body with highest accumulation in the small intestine (44–50%ID/g at 5 min p.i.) and the liver (11–13%ID/g at 5 min p.i.) [123]. The activity levels in all tissues, except the small intestine, decreased to <1%ID/g within 60 min p.i. while relatively low uptake in the kidneys was observed pointing to an elimination of all three radioligands via the enterohepatic circulation [123]. Additionally, defluorination of [^18^F]**39**, [^18^F]**40** and [^18^F]**30** was insignificant as represented by the very low activity accumulation in the bones (≤1%ID/g over 120 min p.i.) [123]. Based on these biodistribution data and whole body dosimetry analysis in mice, the same effective dose of around 13–14 µSv/MBq in a standard human was estimated for [^18^F]**39**, [^18^F]**40** and [^18^F]**30** [123].

Metabolism studies in rats demonstrated low in vivo stability with only 21–24% of intact [^18^F]**39**, [^18^F]**40** and [^18^F]**30** in plasma at 60 min p.i. while two more polar radiometabolites were detected for each radioligand [123]. However, 96% of intact [^18^F]**39** and [^18^F]**40** were observed in the brain at 60 min p.i. indicating that no significant amounts of brain-penetrating radiometabolites were formed. Interestingly, about 17% of one polar radiometabolite of [^18^F]**30** were detected in the brain at 60 min p.i. [123]. It was supposed that this radiometabolite represents [^18^F]fluoroacetate, which is known to enter the brain and to be frequently formed from radioligands bearing a [^18^F]fluoroethoxy group [1,18,74,123,179]. Overall, these findings indicate that [^18^F]**39** and [^18^F]**40** might show an improved metabolic profile compared to [^18^F]**30** in the brain [123]. 

Out of this series of novel PDE10A radioligands, [^18^F]**40** displayed the highest striatal binding in baseline PET scans in rats as discussed above and was thus selected for evaluation of the specificity of binding [123]. In chase-blocking studies, the striatal uptake of both [^18^F]**40** and [^18^F]**30** significantly decreased by 80–90% after administration of **30** (5 mg/kg) at 20 min p.i. of the respective radioligand [123]. Hence, specific binding of both radioligands to PDE10A in the striatum could be confirmed [123]. In conclusion, Mori et al. developed two highly potent PDE10A radioligands, [^18^F]**39** and [^18^F]**40**, which might present superior imaging profiles over [^18^F]**30** for the in vivo assessment of the PDE10A enzyme in the brain by PET. Next, translation of [^18^F]**40** to clinical PET studies and a head-to-head comparison with [^18^F]**30** in humans are planned [123].

At the virtual meeting of the SNMMI in 2020, Cai et al. [178] reported on the development and preliminary biological evaluation of the novel PDE10A radioligand [^18^F]**P10A-1910** ([^18^F]**41**, see Figure 7). The aim of this work was to identify an aryl ^18^F-labeled analog of [^18^F]**30** (Figure 6) that can be prepared by a facile radiosynthetic procedure and displays high brain uptake as well as excellent in vivo stability and target specificity for the assessment of PDE10A in the brain by PET [178]. Unfortunately, no data regarding the inhibitory potency or binding affinity of **41** towards PDE10A were published in that conference abstract. The aromatic ^18^F-labeling was described as a Cu-mediated approach [178], which indicates that the precursor might be a boronic acid or boronic acid ester. In that regard, there are no details given about the radiosynthesis except of a radiochemical yield of 28 ± 3% (not decay corrected) and a molar activity of over 37 GBq/µmol [178].

Biodistribution studies in mice revealed high accumulation of [^18^F]**41** in the small intestine and liver with over 50%ID/g at 5 min p.i. indicating an enterohepatic excretion [178] like described above for the structurally related [^18^F]**30**, [^18^F]**39** and [^18^F]**40** [123]. Metabolism studies exhibited about 80% of intact [^18^F]**41** in the mouse brain at 30 min p.i. [178] pointing to a significant amount of brain-penetrating radiometabolites. Notably, these radiometabolites might be formed faster and to a higher extent compared to the degradation of [^18^F]**30**, [^18^F]**39** and [^18^F]**40** in mice (83–96% of intact radioligand in the brain at 60 min p.i. [123]).

However, the first PET studies in non-human primates showed rapid uptake of [^18^F]**41** in the brain with heterogeneous distribution and high accumulation in the striatum that declined slowly from a SUV of 1.6 at 3 min p.i. to 0.9 at 90 min p.i. [178]. These data indicate that [^18^F]**41** specifically binds to the PDE10A enzyme in the striatum [178]. Overall, Cai et al. suggested [^18^F]**41** as a promising novel PDE10A radioligand that will be further investigated [178].

## 8. Summary and Concluding Remarks

This follow-up review summarizes the extensive efforts of several research groups in the field of molecular imaging of cyclic nucleotide phosphodiesterases by PET since 2016. In fact, a considerable number of pre-clinical and clinical PET studies using already known and well-established radioligands have been published for the PDE isoforms 2A, 4 (B,D) and 10A. 

Starting with the PDE2A enzyme, the radioligand [^18^F]**4** ([^18^F]**PF-05270430**) has been further evaluated to confirm its suitability for target occupancy examinations in the assessment of novel PDE2A inhibitors as therapeutics for neuropsychiatric and neurodegenerative diseases [21]. Although the possible formation of brain-penetrating radiometabolites remains still unclear and might require additional investigations, [^18^F]**4** is the most potent PDE2A-specific radioligand so far. 

Particularly for the isoforms PDE4 (B,D) and PDE10A, a lot of new insights have been reported regarding the complex relationship between altered enzyme density or activity and the pathophysiology of diseases of the central nervous system. For example, current PET studies with the well-known PDE4 radioligand [^11^C]**rolipram** indicated that changes in the PDE4 availability might be associated with cognitive impairments and sleep dysfunctions in Parkinson’s disease (PD) [44,46] as well as with dysregulations in the disrupted in schizophrenia protein 1 (DISC1) related to the progression of neuropsychiatric disorders [51].

With regard to the PDE10A enzyme, in vitro and in vivo binding studies using the radioligand [^18^F]**26** ([^18^F]**JNJ42259152**) revealed important findings about the activation of PDE10A by increased levels of cAMP, which was also observed indirectly via selective inhibition of PDE2A or PDE4 pointing to a strong correlation between different PDE isoforms in the brain [111]. Moreover, PET studies with the PDE10A radioligands [^18^F]**27**, [^18^F]**30** and [^11^C]**31** ([^18^F]**TZ19106B**, [^18^F]**MNI-659** and [^11^C]**IMA107**) provided further evidence that PDE10A is a key enzyme in the regulation of striatal signaling not only in cAMP-dependent but also in dopaminergic pathways. A functional association between PDE10A and the dopamine D_2_ and D_3_ receptors as well as the dopamine transporter in the striatum was shown under physiological conditions [131,140], in HD [143,145] and in PD [155]. Additionally, [^11^C]**31** PET studies indicated that homozygous or heterozygous mutations in the *PDE10A* gene are linked with dysregulation and pathological reduced striatal expression of the enzyme that might lead to the progression of movement disorders [153,154]. Furthermore, preliminary [^11^C]**31** PET data revealed a considerable loss of PDE10A in extra-striatal brain regions, like the insular cortex and occipital fusiform gyrus, of patients with early pre-manifest HD demonstrating that dysregulation of PDE10A is an initial pathophysiological event in the progression of the disease and might be associated with cognitive and behavioral disorders in manifest HD [150].

Besides the intensively studied role of PDE10A in various brain disorders, we have shown that this enzyme is highly expressed in the brown adipose tissue by a retrospective analysis of human [^18^F]**30** PET data and using our PDE10A-specific radioligand [^18^F]**34** ([^18^F]**AQ28A**) in different mouse models of obesity [175,176]. The preliminary results clearly point to a relationship between PDE10A activity and the regulation of energy homeostasis including thermogenic gene expression, lipolysis and glucose uptake suggesting that PDE10A might be a suitable target for the treatment of obesity.

During the last five years, an overall number of 31 novel radioligands have been developed for PET imaging and quantification of PDE1, PDE2A, PDE4 (B, D), PDE5, PDE7 (A, B) and PDE10A (see Table 1).

For PET studies of the respective PDE isoforms in the brain, application of [^11^C]**1**, [^18^F]**2**, [^11^C]**8–12**, [^18^F]**14**, [^11^C]**15**, [^18^F]**16**, [^18^F]**17**, [^18^F]**36** and [^18^F]**38** is hampered due to insufficient brain uptake, poor specific binding, formation of brain-penetrating radiometabolites or unfavorable kinetic profiles. Notably, [^11^C]**1** ((±)-[^11^C]**PF-04822163**) [10] is the only further radioligand that has been developed for brain PET imaging of the PDE1 enzyme since the first PDE1 radioligands have been reported in 2011 [8,9]. Furthermore, the development of a selective PDE4 radioligand that differentiates between the isoforms PDE4B and PDE4D in the brain is gaining interest. In that regard, the novel radioligands [^18^F]**5** for PDE4B [55] and [^11^C]**13** for PDE4D [66] are currently the most promising candidates and under further investigation. Neuroimaging of PDE5 is challenging due to the low enzyme expression in the brain, but the radioligands [^11^C]**19** and [^11^C]**20** showed picomolar potency and excellent selectivity towards PDE5 [75] and thus, in vivo evaluation of these will certainly be followed with great attention. Besides, [^18^F]**16** is proposed as appropriate PDE5 radioligand for the examination of altered enzyme density in heart failure and for myocardial PDE5 occupancy studies [72]. With that regard, first pre-clinical evaluation of the probably more potent PDE5 radioligand [^11^C]**21** ([^11^C]**TPN171**) [77] will be anticipated with high interest. For visualization of the PDE7 enzyme in the brain, only the two novel radioligands [^11^C]**25** ([^11^C]**MTP38**) and [^3^H]**24** have been published since 2016. Pre-clinical and the first-in-human PET studies revealed a promising imaging profile of [^11^C]**25** indicating that this might be the first suitable PDE7 radioligand [99,100]. Furthermore, [^3^H]**24** seems to be a highly potent candidate for the development of a corresponding PET radioligand [98]. Interestingly, all of the novel PDE10A-specific radioligands have been developed based on the molecular structures of the already known radioligands [^11^C]**33** ([^11^C]**T-773**) and [^18^F]**30**. The most promising candidates thereof are [^18^F]**35**, [^18^F]**39** and [^18^F]**40**, which showed superior imaging characteristics compared to the former and well-established analogs.

To the best of our knowledge, out of all the PDE radioligands discussed here the following have already been evaluated in humans: [^18^F]**4** for PDE2A, [^11^C]**rolipram** and [^11^C]**8** for PDE4 (D), [^11^C]**25** for PDE7A/B as well as [^18^F]**26**, [^18^F]**30**, [^11^C]**31**, [^11^C]**32** ([^11^C]**LuAE92686**), and [^11^C]**33** for PDE10A.

In conclusion, the comprehensive work on the investigation of PDEs by PET reviewed here reflects the importance of appropriate isoform-specific radioligands for non-invasive visualization and quantification of this intracellular enzyme class. In future, growing understanding of the functional role of PDEs in the pathophysiology of numerous diseases will certainly set milestones in related clinical research.

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
