# Peer review of "Challenges on Cyclic Nucleotide Phosphodiesterases Imaging with Positron Emission Tomography: Novel Radioligands and (Pre-)Clinical Insights since 2016"

_ijms, 2021, doi:10.3390/ijms22083832_

Round 1
Reviewer 1 Report
This article provides an important update of this emerging area of PET Imaging agents to investigate the roles cyclic nucleotide phosphodiesterases in pharmaceutical research. The authors are to be congratulated on not only reviewing the area but providing valuable insights. I have only minor comments and observations for the authors which may improve the readability of the article. I have noted these below:
Line 35. Insert an between as ... important.
Lines 74-76. This is confusing. I would recommend changing to:
performed by two different methods using (I) tetra-n-butylammonium [18F]fluoride ([18F]TBAF) in tert-amyl alcohol according to Zhang et al. [10] or (II) K+/[18F]F-/K2.2.2-carbonate complex in N,N-dimethylformamide (DMF) as reported by Chen et al. [12]. Morley et al. [15] using dimethyl sulfoxide (DMSO) instead of DMF where able to make a significant improvement in RCY (see Scheme 2).
Line 78. Insert reference 10 in reference list: Scheme 2. Radiosynthesis of [18F]3 [10,12,15].
Line 92. Insert a: Thus, [18F]3 is stated as a suitable
Line 144. Delete the in: ..... brain regions or the altered PDE4 …. (or replace the the with that)
Line 156. Replace ml with mL (consistency and correct usage!)
Line 266. Please check the numbers and times (not sure this makes sense).
Line 287. Delete seven
Line 293. Replace though with thought
Line 349. Insert a in ....is suggested as a suitable PDE5 ...
Line 380. Insert a in …. PDE5 enzyme as a promising non-beta
Line 617. Units: ml vs mL - 31 ng/ml, 75 ng/ml and 99 ng/ml vs 31 ng/mL, 75 ng/mL and 99 ng/mL
Line 624. Units: ml vs mL
Line 632. I would suggest postulated vs presumed.
Line 788. I would suggest postulated vs presumed. (or another synonym)
Line 798. Spelling: replace inconsistence with inconsistency.
Line 917. Replace markedly with marked
Line 950. (Figure 7) should read (Figure 6)
Line 960. Please check this reference I believe it should be 170
Line 969. Replace of with for in ...Except of one lipophilic ...
Line 978. Insert to know in … of interest to know whether
Line 1013. Replace were with was
A bit of tidying up is required in the references
Line 1148. Reference 13: Insert Suppl.#2 and delete the final dash. J. Nucl. Med. 2013, 1147 54, (Suppl.#2) 201
Line 1150. Reference 14: Replace of : Journal of nuclear official publication, Society of Nuclear Medicine with J. Nucl. Med.
Line 1262. Insert (Suppl. #1) into reference - J. Nucl. Med. 2020, 61(Suppl. 1),268
Line 1348. Correct 2020.2010.2029.354696. to 2020.10.29.354696.
Line 1350. Add in press
Line 1380. Remove official journal of the Movement Disorder Society from Movement disorders : official journal of the Movement Disorder Society
Line 1493. Remove full first names of the authors
Line 1499. Remove official journal of the Movement Disorder Society from Movement disorders : official journal of the Movement Disorder Society
Line 1502. Remove official journal of the Movement Disorder Society from Movement disorders : official journal of the Movement Disorder Society
Line 11566. Insert (Suppl. #1) into reference - J. Nucl. Med. 2020, 61(Suppl. 1),1040 AND delete the final -1040.
Finally, I found the use of "besides" used far too often. I would ask that you go through the document and either delete "besides" or replace it with synonym e.g. further; furthermore, additionally; …..
I realize much of this "picky" but I hope it will improve the readability of the manuscript.
Notwithstanding the comments above I greatly appreciate the effort required to bring this together for a very useful document. Thank you.
Author Response
Dear Reviewer,
We would like to thank you for your positive feedback concerning our submitted review entitled: “Challenges on cyclic nucleotide phosphodiesterases imaging with positron emission tomography: Novel radioligands and (pre-)clinical insights since 2016” as an invited “Featured Article” for the special issue “Role of Phosphodiesterase in Biology and Pathology” in the International Journal of Molecular Sciences.
Revisions of the manuscript have been done regarding your helpful comments and suggestions as depicted below. Due to the question of the second Reviewer about the current status of PDE1 radioligands, we checked the literature carefully again and found one corresponding conference abstract only on Research Gate that has been published by Kealey et al. in 2018 (Ref. No 10). Thus, additional changes have been done regarding a new paragraph for PDE1 radioligands (section 2) and the development as well as preliminary biological evaluation of (±)-[11C]PF-04822163 (review No. [11C]1, see section 2), including a re-numbering of all following compounds in the text, Figures, Schemes, and Table of sections 3 to 8. Any revisions are clearly highlighted using the "Track Changes" function in Microsoft Word.
Best regards,
Matthias Scheunemann
Line 35. Insert an between as ... important. ïƒ This has been changed.
Lines 74-76. This is confusing. I would recommend changing to:
performed by two different methods using (I) tetra-n-butylammonium [18F]fluoride ([18F]TBAF) in tert-amyl alcohol according to Zhang et al. [10] or (II) K+/[18F]F-/K2.2.2-carbonate complex in N,N-dimethylformamide (DMF) as reported by Chen et al. [12]. Morley et al. [15] using dimethyl sulfoxide (DMSO) instead of DMF where able to make a significant improvement in RCY (see Scheme 2). ïƒ Thank you very much for pointing that out! It has been changed to “Chen et al. performed the radiosynthesis of [18F]4 by two different methods using (I) tetra-n-butylammonium [18F]fluoride ([18F]TBAF) in tert-amyl alcohol according to Zhang et al. [19] or (II) K+/[18F]F-/K2.2.2-carbonate complex as described by Morley et al. [24] but in N,N-dimethylformamide (DMF) instead of dimethyl sulfoxide (DMSO) [21] (see Scheme 3).”. We hope it is clear now – Chen et al. used both methods and we added the results from Zhang et al. in Scheme 3 for direct comparison.
Line 78. Insert reference 10 in reference list: Scheme 2. Radiosynthesis of [18F]3 [10,12,15]. ïƒ This has been changed (Scheme 3. Radiosynthesis of [18F]4 [19,21,24]).
Line 92. Insert a: Thus, [18F]3 is stated as a suitable ïƒ This has been changed.
Line 144. Delete the in: ..... brain regions or the altered PDE4 …. (or replace the the with that) ïƒ This has been changed.
Line 156. Replace ml with mL (consistency and correct usage!) ïƒ This has been changed.
Line 266. Please check the numbers and times (not sure this makes sense). ïƒ Times and numbers were correct, but we deleted the slow decline of the SUVs from 60 to 120 min, because it seems not important. Thus, we changed that to “In the human brain, a high uptake with peak SUV of 4.39 at 4 min p.i. was observed that slowly declined and plateaued from 30 min to 120 min (SUV ~ 2).”
Line 287. Delete seven ïƒ This has been changed.
Line 293. Replace though with thought ïƒ This has been changed.
Line 349. Insert a in ....is suggested as a suitable PDE5 ... ïƒ This has been changed.
Line 380. Insert a in …. PDE5 enzyme as a promising non-beta ïƒ This has been changed.
Line 617. Units: ml vs mL - 31 ng/ml, 75 ng/ml and 99 ng/ml vs 31 ng/mL, 75 ng/mL and 99 ng/mL ïƒ This has been changed.
Line 624. Units: ml vs mL ïƒ This has been changed.
Line 632. I would suggest postulated vs presumed. ïƒ This has been changed.
Line 788. I would suggest postulated vs presumed. (or another synonym) ïƒ This has been changed to “assumed”.
Line 798. Spelling: replace inconsistence with inconsistency. ïƒ This has been changed.
Line 917. Replace markedly with marked ïƒ This has been changed.
Line 950. (Figure 7) should read (Figure 6) ïƒ This has been changed to (Figures 6 and 7) – the molecular structures and IC50 values of compound 32 (now 33) are shown in Figure 6 and the respective data for compounds 34 (now 35) and 35 (now 36) are given in Figure 7.
Line 960. Please check this reference I believe it should be 170 ïƒ Thanks a lot for this important hint! This has been changed (now Ref. 177).
Line 969. Replace of with for in ...Except of one lipophilic ... ïƒ This has been changed.
Line 978. Insert to know in … of interest to know whether ïƒ This has been changed.
Line 1013. Replace were with was ïƒ We suggest that “96% of … were observed” is correct.
A bit of tidying up is required in the references
Line 1148. Reference 13: Insert Suppl.#2 and delete the final dash. J. Nucl. Med. 2013, 1147 54, (Suppl.#2) 201 ïƒ This has been changed.
Line 1150. Reference 14: Replace of : Journal of nuclear official publication, Society of Nuclear Medicine with J. Nucl. Med. ïƒ This has been changed.
Line 1262. Insert (Suppl. #1) into reference - J. Nucl. Med. 2020, 61(Suppl. 1),268 ïƒ This has been changed.
Line 1348. Correct 2020.2010.2029.354696. to 2020.10.29.354696. ïƒ This has been changed.
Line 1350. Add in press ïƒ This has been added.
Line 1380. Remove official journal of the Movement Disorder Society from Movement disorders : official journal of the Movement Disorder Society ïƒ This has been changed.
Line 1493. Remove full first names of the authors ïƒ This has been changed.
Line 1499. Remove official journal of the Movement Disorder Society from Movement disorders : official journal of the Movement Disorder Society ïƒ This has been changed.
Line 1502. Remove official journal of the Movement Disorder Society from Movement disorders : official journal of the Movement Disorder Society ïƒ This has been changed.
Line 11566. Insert (Suppl. #1) into reference - J. Nucl. Med. 2020, 61(Suppl. 1),1040 AND delete the final -1040. ïƒ This has been changed.
Finally, I found the use of "besides" used far too often. I would ask that you go through the document and either delete "besides" or replace it with synonym e.g. further; furthermore, additionally; ….. ïƒ "Besides" has been replaced four times (3 x Moreover, 1 x In addition).
I realize much of this "picky" but I hope it will improve the readability of the manuscript. ïƒ We are completely fine with that and appreciate every comment that helps to improve our manuscript!

Reviewer 2 Report
This is a very nice comprehensive and timely state-of-the-art review on the use of phosphodiesterase ligands for PET. Lots of new information and chemical structures have been included since the last version of such a review by the authors published in 2016 (ref 1). Mostly in the filed of PDE4 and PDE10 inhibitors lots of progress has been made in the last years and these developments are highlighted in more detail, especially PDE10. The paper is very well written and will be of high interest to researchers in the field.
I have three minor suggestions we should be addressed before publication:
- There is a discrepancy in the nomenclature of PDE enzyme superfamily. In the introduction, lines 28 and 37 the authors use the term "family subtypes" which is odd. The commonly used nomenclature is: PDE superfamily - PDE families (PDE1, 2, 3, 4, etc), PDE subfamilies (e.f. PDE4A, PDE4B, 4C, 4D) and PDE isoforms (PDE4D3, PDE4D5 etc). Please correct. Ironically, correct nomenclature was used in their 2006 review.
- Are there any developments in the field for PDE1 ligands at all? on line 37 it is not said that no ligands for PDE1 have been developed
- As per journal style- should full chemical names be included somewhere, at least as a list in supplement to fully name/describe compounds under discussion in addition to 1, 2, 3, etc used in the main text?
Author Response
Dear Reviewer,
We would like to thank you for your positive feedback concerning our submitted review entitled: “Challenges on cyclic nucleotide phosphodiesterases imaging with positron emission tomography: Novel radioligands and (pre-)clinical insights since 2016” as an invited “Featured Article” for the special issue “Role of Phosphodiesterase in Biology and Pathology” in the International Journal of Molecular Sciences.
Revisions of the manuscript have been done regarding your helpful comments and suggestions as depicted below. Due to your question about the current status of PDE1 radioligands, we checked the literature carefully again and found one corresponding conference abstract only on Research Gate that has been published by Kealey et al. in 2018 (Ref. No 10). Thus, additional changes have been done regarding a new paragraph for PDE1 radioligands (section 2) and the development as well as preliminary biological evaluation of (±)-[11C]PF-04822163 (review No. [11C]1, see section 2), including a re-numbering of all following compounds in the text, Figures, Schemes, and Table of sections 3 to 8. Any revisions are clearly highlighted using the "Track Changes" function in Microsoft Word.
Best regards,
Matthias Scheunemann
There is a discrepancy in the nomenclature of PDE enzyme superfamily. In the introduction, lines 28 and 37 the authors use the term "family subtypes" which is odd. The commonly used nomenclature is: PDE superfamily - PDE families (PDE1, 2, 3, 4, etc), PDE subfamilies (e.f. PDE4A, PDE4B, 4C, 4D) and PDE isoforms (PDE4D3, PDE4D5 etc). Please correct. Ironically, correct nomenclature was used in their 2006 review. ïƒ Thanks for that important hint! It has been changed.
Are there any developments in the field for PDE1 ligands at all? on line 37 it is not said that no ligands for PDE1 have been developed ïƒ This is an interesting question! In 2011, the first potential PDE1 radioligands have been claimed in patent by Li et al. and as mentioned above, we checked the literature again and added a new paragraph regarding PDE1 radioligands. We thank you very much for pointing that out!
As per journal style- should full chemical names be included somewhere, at least as a list in supplement to fully name/describe compounds under discussion in addition to 1, 2, 3, etc used in the main text? ïƒ Since this is a review article, we would not like to provide a supplement. The molecular structures (if known/published) and references of all the discussed compounds are given. Thus, we suggest that it is not necessary to provide the IUPAC names. If mentioned in the main text, this would impair the readability of the manuscript from our point of view. We hope you agree with that point!
